# Anomalous distribution of distinctive water masses over the Carlsberg Ridge in May 2012

Hailun He[1], Yuan Wang[2], Xiqiu Han[2], Yanzhou Wei[1], Pengfei Lin[3], Zhongyan Qiu[2], and Yejian Wang[2]

[1]State Key Laboratory of Satellite Ocean Environment Dynamics, Second Institute of Oceanography, Ministry of Natural Resources, Hangzhou 310012, China

[2]Key Laboratory of Submarine Geosciences, Second Institute of Oceanography, Ministry of Natural Resources, Hangzhou 310012, China

[3]State Key Laboratory of Numerical Modeling for Atmospheric Sciences and Geophysical Fluid Dynamics, Institute of Atmospheric Physics, Chinese Academy of Sciences, Beijing 100029, China

**Correspondence:** Hailun He (hehailun@sio.org.cn); Xiqiu Han (xqhan@sio.org.cn)

**Abstract.** In May 2012, we conducted a hydrographic survey over the Carlsberg Ridge in the northwest Indian Ocean. In this paper, we use these station data, in combination with some free-floating Argo profiles, to obtain the sectional temperature and salinity fields, and subsequently, the hydrographic characteristics are comprehensively analyzed. Through the basic T-S diagram, three salty water masses, Arabian Sea High-Salinity Water, Persian Gulf Water, and Red Sea Water, are identified. The sectional data show a clear ventilation structure associated with Arabian Sea High-Salinity Water. The 35.8 psu salty water sinks at 6.9°N and extends southward to 4.4°N at depths around the thermocline, where the thermocline depth is in the range of 100 to 150 m. This salty thermocline extends much further south than the climatology indicates. Furthermore, the temperature and salinity data are used to compute the absolute geostrophic current over the specific section, and the results show meso-scale eddy vertical structure different from some widely used oceanic reanalysis data. We also find a west-propagating planetary wave at 6°N, and the related features are described in terms of phase speed, horizontal and vertical structures.

## 1   Introduction

The northwest Indian Ocean (NWIO) is unique compared with the other two basin-scale oceans (Pacific and Atlantic Oceans) because the dominant characteristics are monsoon driven (Schott and McCreary Jr., 2001; Schott et al., 2009). The seasonal monsoon forces the coastal current back and forth and generates the Somalia Current, which is always marked as the strongest current in the real ocean (as strong as 3.5 m s$^{-1}$). Moreover, the monsoon is strong enough to change the pattern of basin-scale circulation. The monsoon builds up a dominant meridional current in the NWIO, changing the form of the customary zonal current (as in the Pacific and Atlantic Oceans) into a meridional current. NWIO is also famous for its role in the so-called Indian Ocean Dipole (Saji et al., 1999; Webster et al., 1999; Han et al., 2014; Chen et al., 2015), which represents the zonal gradient of sea surface temperature in the Indian Ocean. As a basin-wide signal, the Indian Ocean Dipole is closely related to the Indian-Ocean-adjacent climate (Li and Han, 2015). Some studies also emphasized the distinct meso- and submeso-scale air-sea interactions in the NWIO (Vecchi et al., 2004).

To date, the main water masses in the NWIO have been described by the scientific community (Sharma et al., 1978; Kumar and Prasad, 1999; Emery, 2001; Talley et al., 2011). For instance, three water masses were defined in NWIO as Arabian Sea High-Salinity Water (ASHSW), Persian Gulf Water (PGW) and Red Sea Water (RSW). The formations of ASHSW, PGW and RSW are all due to the high evaporation (Shapiro and Meschanov, 1991; Kumar and Prasad, 1999; Bower et al., 2000; Prasad
et al., 2001; Prasad and Ikeda, 2002). Regarding the pathways of these water masses, the movements are not well observed, and the corresponding dynamics are not clear. Kumar and Prasad (1999) described the climatological seasonal distribution of ASHSW using in-situ temperature and salinity fields. In the northern Arabian Sea, ASHSW forms in the surface during winter, and moves southward due to the surface wind. Otherwise, the multi-scale variations of ASHSW were not sufficiently documented (Kumar and Prasad, 1999; Prasad and Ikeda, 2002). According to the customary ocean ventilation theory, ASHSW
sinks and moves southward along the isopycnal layer from the generation zone following the wind-driven current (Luyten et al., 1983). However, the applicability of the ocean ventilation theory is still unknown for the NWIO, because the surface wind reverses direction under the influence of winter monsoon (Liu et al., 2018). RSW supplies important intermediate water salinity source in the entire Indian Ocean basin (Han and McCreary Jr., 2001). Formation and spreading of RSW exhibit seasonal variations (Bower et al., 2000; Beal et al., 2000). Based on the long-term hydrographic data, the occurrence of RSW
show four possible branches of RSW around the Gulf of Aden: First spreads southward along the Somali coast, the second moves southward to Somali Basin, the third flows eastward to the Arabian Basin, and the fourth moves northeastward along the Arabian coast (Shapiro and Meschanov, 1991). Later, Beal et al. (2000) highlighted the branch along the Somali coast, meanwhile, the potential vorticity analysis revealed that the flows generally followed the zonal direction in NWIO. It is noted that the dynamics of intermediate water in Indian Ocean should be different from that in the Pacific and Atlantic Oceans. The
limited meridional extent of Indian Ocean omits the polar-to-subpolar front, which helps forming intermediate water in Pacific and Atlantic Oceans (You, 1998). How the RSW moves is worthy further investigation (Durgadoo et al., 2017).

On the other hand, the meso-scale eddies and planetary waves are not sufficiently observed in NWIO. The historical and present Research Moored Array for African-Asian-Australian Monsoon Analysis and Prediction observation arrays are close to the equator and omit the NWIO. In situ observations in the NWIO mainly depend on Array for Real-time Geostrophic
Oceanography (Argo; Riser et al., 2016; Vitale et al., 2017). However, as we show later, the number of Argo floats is still too sparse to represent the meso-scale eddy field in the NWIO. Besides, the planetary waves at-least include Rossby and Kelvin waves (Rhines, 1975; McCreary, 1985). Satellite-retrieved sea surface height are commonly used to detect the planetary waves (Chelton and Schlax, 1996), while the internal dynamics of planetary wave is not sufficiently addressed. The phase speed of west-propagating planetary waves (WPPW) has been matched to the theoretical Rossby wave (Chelton and Schlax, 1996; Sub-
rahmanyam et al., 2001), nonetheless, the vertical structure of WPPW calls for vertical profiling observation (Subrahmanyam et al., 2001).

The present circumstance stimulates our effort to find more observational resources. The Carlsberg Ridge (CR) is a typical slow-spreading ridge and lies along the northwest-southeast direction in the NWIO. Recently, we conducted an interdisciplinary survey on CR (Yang et al., 2016; Wang et al., 2017), and the hydrographic analysis of CR is necessary for at least three reasons.
First, hydrographic survey takes a snapshot on the water mass, and gives an evidence on the movement of water mass. Second,

the observation probably captures the vertical structure of meso-scale eddy or planetary wave. Third, the results could be used to evaluate the widely-used oceanic reanalysis. The trajectory of Argo float are not manually controlled; however, ship surveys could cover specified sections and have a clearer objective. Hence, this paper is motivated by observing the water masses over CR, describing the vertical structures of meso-scale eddy and planetary wave, and comparing the results of widely-used oceanic reanalysis in the NWIO.

## 2 Data and methods

### 2.1 In situ data description

The data for our study were collected during the Chinese cruise DY125-24 (May 2012) by the Chinese research vessel "LISIGUANG". Hydrographic observations were conducted in the region of the CR. The vertical profiles of temperature, conductivity and pressure were obtained by a calibrated SBE-19plus CTD and some expendable CTD (XCTD). The station information is shown in Fig. 1 and Table 1. All stations were mainly located close to the CR section and therefore defined the regional along-section ($y$) and cross-section ($x$) coordinates. The maximum measurement depths of XCTD and Argo are 1050 and 2000 m, and therefore we limited our analysis to depth 2000 m. According to Talley et al. (2011) and Emery (2001), upper 2000 m depth covers the upper ocean (defined as 0-500 m) and the intermediate-depth ocean (defined as 500-2000 m). In our postprocessing, 13 simultaneous Argo profiles were found within a 200 km radius of the study region (Fig. 2 and Table 1).

Regarding data quality, an intercomparison between XCTD and CTD measurement was implemented in the southern tropical Indian Ocean. The XCTD station involved was located at 73.8°E and 1.7°S at 14:23 on May 4 (Coordinated Universal Time), and the counterpart CTD station was located at 73.5°E and 1.4°S a half-hour later (14:58 on May 4; distance with the XCTD station as 47.17 km). The mean differences in the recorded in situ temperature (salinity) were 0.425°C (0.058 psu) in the upper ocean and 0.051°C (0.053 psu) in the intermediate-depth ocean. It is noted that the differences among CTD and XCTD are not negligible. However, because the distance between two stations is relatively large, and the bias of XCTD is different for different instruments, we does not perform instrument calibration. We later use objective analysis method and low-pass filter to smooth the data among CTD, XCTD and Argo.

### 2.2 In situ data processing

All the data from several sources need to be processed to same levels because of the different sampling rates; i.e., the vertical resolutions of CTD, XCTD and Argo are 0.1, 0.1, and 2.0 m, respectively. In the first step of data postprocessing, the coarse data are moving-averaged into a uniform vertical grid with a 5 m interval starting from 5 m below the surface. Here 5 m vertical resolution is sufficient for describing vertical structure of mixed-layer and water masses. Special treatment is imposed on one Argo float (2901888; three profiles; Table 1), where the coarse profiles lose data in the upper 20 m; thus, the missing data are filled with the same value as uppermost available data in the near surface.

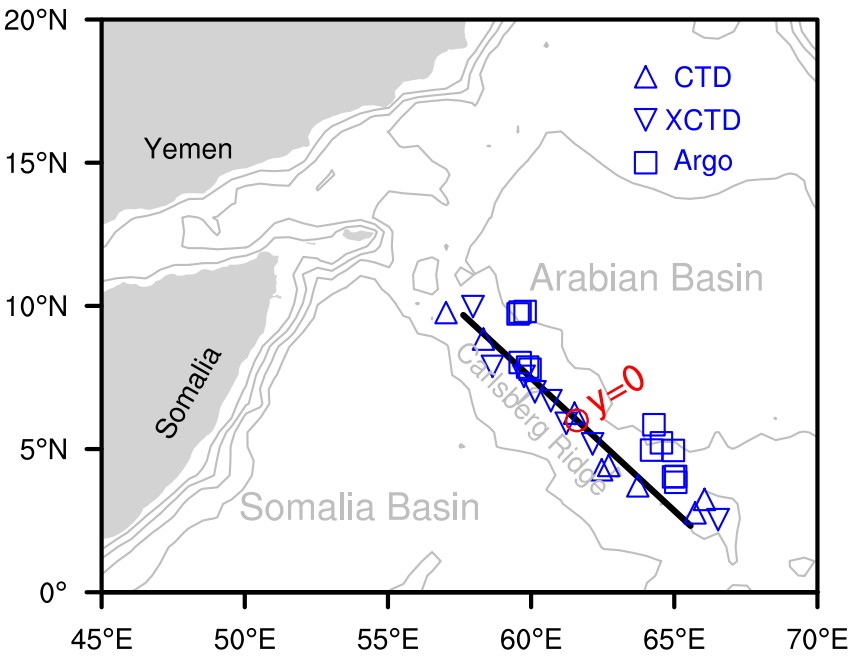

**Figure 1.** CTD/XCTD stations of the DY125-24 survey and the simultaneous Argo profiles around the CR. The CR defines local coordinates in which the $x$-coordinate is cross-track and the $y$-coordinate along-track, the corresponding origin point is selected as (61.6°E, 6.0°N), and the isobaths of -4000, -3000, -2000 and -1000 m are presented.

The data are then projected into the standard CR section ($y$-coordinate), and the corresponding grid interval is 50 km. We use the objective analysis method to interpolate data from irregularly spaced locations to a fixed grid (Barnes, 1994). Later, a low-pass filter is imposed on the CR sectional data to remove the short-wavelength signals, which are partly from the cross-bias among different data sources and partly from the submesoscale or higher wavenumber signals in the real ocean. The low-pass filter is a two-dimensional LOcally Estimated Scatterplot Smoothing (LOESS) filter (Cleveland and Grosse, 1991), and the moving-average wavelengths are 300 km and 30 m in the horizontal and vertical directions, respectively. As a result, the smoothed data save the essential features of the thermal-salinity field but remove the noise.

## 2.3 Satellite data description

### 2.3.1 Surface wind

We use Cross-Calibrated Multi-Platform (CCMP;  Atlas et al., 2011) gridded surface vector winds here (version 2.0). CCMP data are daily products, and they are projected on 0.25°×0.25° grids.

**Table 1.** Information on CTD/XCTD stations and Argo floats.

| Type | Station/Float | Latitude | Longitude | Time (month/day hour) |
|------|---------------|----------|-----------|----------------------|
| XCTD | S09CTD08 | 2.3°N | 66.3°E | 05/06 07 |
| XCTD | S16CTD13 | 5.1°N | 62.1°E | 05/14 16 |
| XCTD | S19CTD15 | 6.4°N | 60.4°E | 05/16 13 |
| XCTD | S20CTD16 | 7.3°N | 59.5°E | 05/16 19 |
| XCTD | S23CTD19 | 7.0°N | 60.1°E | 05/20 14 |
| XCTD | S25CTD21 | 9.6°N | 57.6°E | 05/23 04 |
| XCTD | S26CTD22 | 5.5°N | 61.1°E | 05/26 20 |
| CTD | S10CTD09 | 3.1°N | 66.0°E | 05/07 05 |
| CTD | S12CTD10 | 2.5°N | 65.4°E | 05/09 12 |
| CTD | S14CTD11 | 4.2°N | 62.3°E | 05/11 17 |
| CTD | S15CTD12 | 4.3°N | 62.4°E | 05/13 21 |
| CTD | S17CTD14 | 6.2°N | 61.3°E | 05/15 08 |
| CTD | S21CTD17 | 8.5°N | 58.2°E | 05/17 05 |
| CTD | S22CTD18 | 9.5°N | 57.0°E | 05/18 07 |
| CTD | S28CTD24 | 3.4°N | 63.4°E | 06/01 04 |
| Argo | 2901847 | 4.0°N | 65.0°E | 05/03 02 |
| Argo | 2901848 | 5.8°N | 64.3°E | 05/07 13 |
| Argo | 2900877 | 8.0°N | 59.6°E | 05/07 09 |
| Argo | 2901096 | 5.0°N | 64.2°E | 05/08 19 |
| Argo | 2901888 | 9.7°N | 59.5°E | 05/08 01 |
| Argo | 2901847 | 4.0°N | 65.0°E | 05/12 20 |
| Argo | 2901888 | 9.8°N | 59.6°E | 05/17 23 |
| Argo | 2900877 | 7.9°N | 59.9°E | 05/17 09 |
| Argo | 2901096 | 5.2°N | 64.6°E | 05/18 19 |
| Argo | 2901847 | 3.8°N | 65.1°E | 05/22 16 |
| Argo | 2900877 | 7.8°N | 60.0°E | 05/27 09 |
| Argo | 2901096 | 5.0°N | 65.0°E | 05/28 18 |
| Argo | 2901888 | 9.8°N | 59.8°E | 05/28 01 |

### 2.3.2 Sea surface temperature

The sea surface temperature (SST) data is produced by Operational Sea Surface Temperature and Sea Ice Analysis (OSTIA; Donlon et al., 2011), which merges satellite infrared and microwave products, ship, buoy, etc. OSTIA is daily product, and the horizontal resolutions are 0.05°×0.05°.

### 2.3.3 Sea surface height

For describing the sea surface height (SSH) and the related surface geostrophic current, we use the Archiving, Validation, and Interpretation of Satellite Oceanographic (AVISO) grided data. The temporal resolution is daily, and the horizontal resolutions are $0.25^{\circ} \times 0.25^{\circ}$.

### 2.4 Reanalysis data description

As references, we also employ two widely used reanalysis datasets for comparison, aiming at evaluating the quality of reanalysis data. The first reanalysis dataset is Simple Ocean Data Assimilation (SODA, version 3.3.1; Carton and Giese, 2008; Carton et al., 2018), whose horizontal resolutions are $0.25^{\circ} \times 0.25^{\circ}$ for longitude and latitude. The second reanalysis dataset is HYbrid Coordinate Ocean Model (HYCOM, version GOPS3.0:HYCOM+NCODA global 1/12$^{\circ}$ Reanalysis GLBu0.08/expt_19.1), and its horizontal resolutions are $0.08^{\circ} \times 0.08^{\circ}$. Both SODA and HYCOM assimilate various in situ and satellite-based data sources: historical station profiles, Argo profiles, moorings, drifters, satellite SSTs, SSHs, etc. For comparison, we extract the reanalysis datasets along the same section as the observations, and the monthly mean fields in May 2012 are used.

### 2.5 Method of tracers

Using SODA reanalysis, we release some passive tracers along the CR and backtrack their trajectories based on the Lagrangian description, and the methods are formulated by

$$
\begin{cases}
X^{n-1/2} & = X^{n+1/2} - U^n \cdot \Delta t \\
Y^{n-1/2} & = Y^{n+1/2} - V^n \cdot \Delta t \\
z^{n-1/2} & = z^{n+1/2} - w^n \cdot \Delta t
\end{cases}
\tag{1}
$$

Here, $X$ and $Y$ are the Cartesian coordinates along longitude and latitude respectively, $U$ and $V$ are the corresponding currents, $z$ is the vertical coordinate, $w$ is the vertical velocity, and $n$ is the time step. In the computation, we use the three dimensional velocity ($U$, $V$, $w$) to track the tracers, and we set the time step ($\Delta t$) as 3600 s. The tracers are set along the CR on May 15, 2012, and then backward integrated to January 1, 2010.

## 3 Results

### 3.1 Background environment

The time period of the shipboard survey starts from 2012/05/06 and ends at 2012/06/01 (Table 1). Figure 2(a-c) show the monthly mean surface wind, SST and SSH, respectively. In this specific month, the summer monsoon has started but is not very strong (Fig. 2a) . The along-coast wind prevails in the regional wind field, and the wind speed in the region far from the western coast is weaker. Positive wind curl along the Somalia coast and Yemen induces coastal upwelling, which brings lower-layer cold water upward and cools the sea surface (Fig. 2b). The patterns of wind curl are roughly consistent with that of

the climatological monthly mean wind stress curl (Beal et al., 2013). Otherwise, the wind curl in the NWIO is negative, which is consistent with the annual mean, and forces downwelling. The basin-scale semicircular SST contour (30°C) then outlines a warm area in the oceanic interior. The main part of the CR is located in this strikingly warm region (Fig. 2b).

On the other hand, SSH (or absolute dynamic height) shows multiple meso-scale eddies (Fig. 2c). There are some warm-core eddies (anti-cyclonic eddies), to the east of the CR (WCE1), east of the Horn of Africa (WCE2), and northeast of the Horn of Africa but very close to Yemen (WCE3). Two cold-core eddies (cyclonic eddies; CCE1 and CCE2) are also observed at either end of the CR. Besides, a remarkable westward current is observed at the latitude of 6°N, which is noted here as a WPPW. WPPW is pronounced compared with the circumstances around the specific region with respect to its zonal extent (7.5° in longitude), while the meridional extent is relatively narrow (1.3° in latitude). The magnitude of zonal current is 0.38 m/s. Furthermore, we investigated the longitude-time plot of the surface zonal current at 6°N latitude (Subrahmanyam et al., 2001, surface geostrophic current from SSH, results are not shown here), the WPPW is observed to start at 69°E on day 102, the surface current propagates westward with a phase speed of 0.2 m s$^{-1}$ and arrives at 60°E on day 155.

## 3.2   Temperature, salinity and density

First, we impose the water mass analysis on the objective-analysis data (section 2.2), and the results are shown in Fig. 3. The data support that the upper water is more saline than the Indian Equatorial Water and fresher than the Arabian Sea Water. The observed waters are likely to be mixed Indian Equatorial Water and Arabian Sea Water. When the latitude spans from the equatorial band (2.3-5°N in the present study) to the tropical band (5-9.7°N in the present study), the salinity generally increases, consistent with the northern-side Arabian Sea Water being much more saline than the southern-side Indian Equatorial Water (Han et al., 2014), and this meridional variation in salinity is due to the different proportions of Indian Equatorial Water and Arabian Sea Water. On the northwest side, water columns contain ASHSW, which are observed as saline water at a potential density of approximately 24 kg/m$^3$ (Kumar and Prasad, 1999).

The intermediate waters from our data are projected as PGW (Prasad et al., 2001) and RSW (Beal et al., 2000; Talley et al., 2011). According to Kumar and Prasad (1999), the definition of PGW (RSW) is the density range as 26.2-26.8 kg/m$^3$ (27.0-27.4 kg/m$^3$), temperature range 13-19°C (9-11°C), and the salinity range 35.1-37.9 psu (35.1-35.7 psu).

Sectional profiles of temperature and salinity are shown in Fig. 4. The thermocline is in the depth range of 100 to 150 m (20°C isothermal line, Xie et al., 2002). From the present observation, the thermocline is nearly flat at the equatorial band and deepens northward in the tropical band. This phenomenon is also supported by the climatological data, which reveal that the sectional distribution of the thermocline is similar to a long-standing geostrophic balanced signal. In the near surface, the isothermal line of 30°C rises to the surface on the northern side suggesting a ventilation structure such that subsurface water can take part in the air-sea interaction. Meanwhile, for the intermediate water, the isothermal line tilts deeper from south to north.

The striking feature of the salinity field is that a salinity tongue appears at 100 m depth, where the salty water is ASHSW (Kumar and Prasad, 1999). Climatological data show that these salty waters originate from the north side and extend southward; however, in our survey, the extent is greater. We emphasize the iso-salinity line of 35.8 psu; the southern extent can reach $y$=150

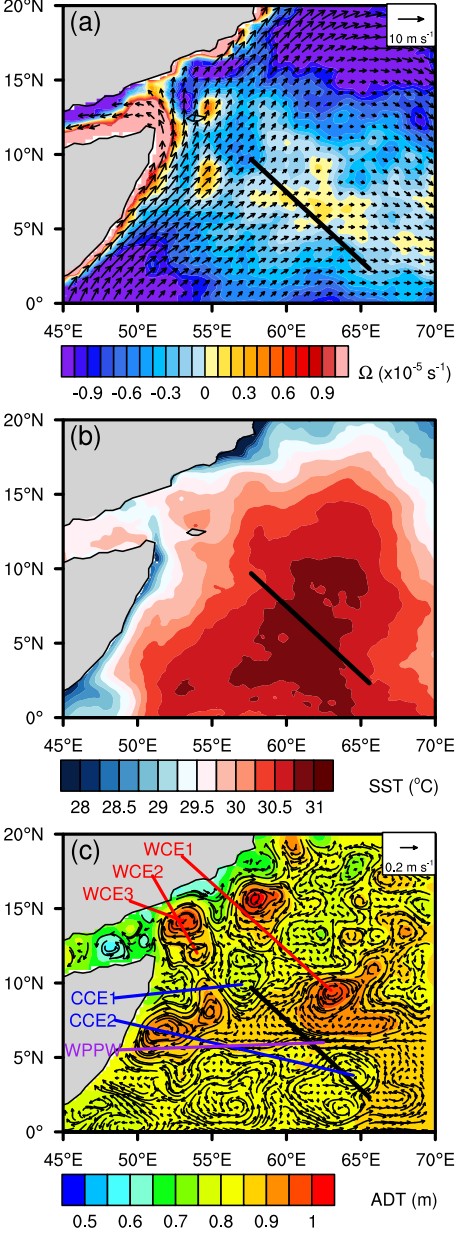

**Figure 2.** (a) Monthly mean wind vector and vorticity (Ω) from CCMP wind data, data are plotted every 3 points for wind vector; (b) monthly mean sea surface temperature from OSTIA data; and (c) monthly mean sea surface height (absolute dynamic topography), and the corresponding surface geostrophic current (shown with every 2 points) from AVISO. The month of May 2012 is considered.

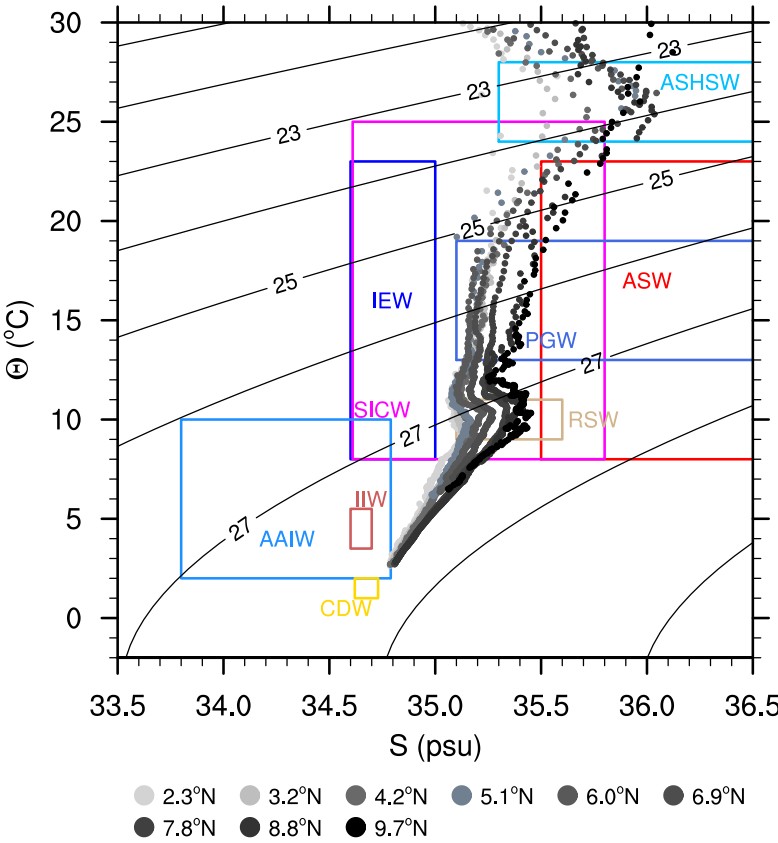

**Figure 3.** T-S diagram for the upper 2000 m of water on the CR. The water masses are defined in Emery (2001) and include the Arabian Sea water (ASW), Indian equatorial water (IEW), South Indian central water (SICW), Antarctic intermediate water (AAIW), Indonesian intermediate water (IIW), and circumpolar deep water (CDW). The rectangles represent the appropriate temperature and salinity ranges (Table 1 in Emery (2001)). Besides, Red Sea water (RSW), Persian Gulf water (PGW), and Arabian Sea High-Salinity Water (ASHSW) are also represented according to Kumar and Prasad (1999).

km (or 6.9°N) in climatology but -250 km (or 4.4°N) in our survey. This result means that the salty water extends southward more than 2.5° in latitude in excess of climatology. The present sectional observation also shows the salty intermediate water as PGW and RSW (Fig. 3). The observation shows slightly more saline water than climatology on the northern side, although

the overall structure is mostly consistent with the climatology. In other words, the comparison implies that the year 2012 is an anomaly year for the distributions of ASHSW, PGW and RSW.

In the potential density field, the appearance of the 22 kg/m$^3$ isopycnal is evident in both the snapshot and climatology. For the snapshot, the outcrop point of the 22 kg/m$^3$ isopycnal is $y$=300 km, or 7.8°N. It is worth noting that the outcrop point is
near the transition point for signs of wind vorticity (Fig. 2a), and the results suggest that the ventilation is induced by surface wind. The north side of the outcrop point has negative wind vorticity, which promotes downward movement. Ventilation is highly related to the sinking of high-salinity water and its southern extent (Luyten et al., 1983; Kumar and Prasad, 1999).

Fig. 4 also shows the reanalysis data, and essentially, the reanalysis captures the thermal structure quite well compared with the present observations and climatology. For instance, in the upper ocean, the surface warm water is distributed on the south
side and northward to $y$=400 km where the isothermal line of 30°C outcrops. The observed descending thermocline near $y$=100 km is recognizable in SODA and HYCOM. It is also noted that the equatorial near-surface upwelling in HYCOM has some evidence from the present observations. For the intermediate-depth water, in SODA and HYCOM, the isothermal lines also tilt deeper from south to north, which resembles both the present observations and the climatology.

For salinity, the southward extent of ASHSW is also captured by SODA and HYCOM, and the southward extent in HYCOM
approaches the observations more closely. In the intermediate-depth water, the southward extent from the north side in SODA is similar to the observations, while the corresponding signal in HYCOM is obscured.

The upper ocean density fields from SODA and HYCOM also show clear ventilation structures, that the south-side subsurface layers are exposed to the surface in the north-side. From the observations, equatorial waters with a potential density of 22 kg/m$^3$ at a depth of 30 m are rising to the surface. The outcrop points of potential density of 22 kg/m$^3$ in SODA and HY-
COM are shifted southward compared with the observations. Additionally, the near-surface upwelling in the equatorial band in HYCOM is strong, but not significant in the observations.

### 3.3   Cross-track current

The observation-based absolute geostrophic current is deduced from the in situ density field by thermal wind theory (Fig. 5), where the velocity is integrated downward from the surface geostrophic current (Lagerloef et al., 1999). The velocity field is
relatively strong in the upper ocean, where the current field is dominated by meso-scale eddies. The cross-track current in the equatorial band is induced by CCE2. The structure of CCE2 is asymmetric, and the positive cross-track flow is stronger than the negative counterpart. In contrast, the northwest side of CR is identified as the south part of CCE1 (northeastward flow). At 6°N latitude, the vertical structure of WPPW is represented in Fig. 5a. WPPW seems to extend vertically to a depth of 200 m, and the horizontal extent is near 200 km for the current greater than 0.02 m/s. Meanwhile, the maximum cross-track current
of the WPPW is 0.12 m/s. The current field also captures the northeastward current (less than 0.075 m/s) in the intermediate depth (-200 $\leq y \leq$ -20 km, and 150 $\leq y \leq$ 350 km), which is due to the corresponding isothermal tilting (Fig. 4).

For the reanalysis data, as shown in Fig. 5, although the surface currents are similar due to the assimilation of SSH in the reanalysis process, the cross-track current from reanalysis is quite different from the observation-based absolute geostrophic current. The differences are observed in three aspects. First, the meso-scale eddy CCE2 is not well represented for the vertical

structure, as SODA and HYCOM limit the southern part of CCE2 to the upper 200 m, where the current speed is faster than 0.05 m/s. Meanwhile, in SODA and HYCOM, the northern part of the meso-scale eddy (CCE2) has much latitude expansion, and merges with the WPPW. Second, the undercurrent in the southern portion of the observations differs from those in SODA and HYCOM. The undercurrent in SODA is relatively weak, while HYCOM shows a northward shift of the current core. Finally, for the northern portion, SODA gives a relatively shallow depth for the surface northeastward current, and the corresponding horizontal extent exceeds that of the observations. The locations of surface zero current in SODA and observation are $y$=170 and 240 km, respectively.

Part of the difference between observation-based absolute geostrophic current and reanalysis current is probably due to the near-surface Ekman current. The mean surface Ekman speed, which is approximately the difference between surface geostrophic current and in-situ surface current (from surface drifter), is within 0.1 m/s in northern Indian Ocean (Saj, 2017). Besides, the climatological monthly mean mixed-layer depth in May is roughly 20 m at station 9.5°N and 59.5°E (on CR; Liu et al., 2018). Considering the near-surface Ekman current decays exponentially with depth, therefore, the near-surface Ekman currents in reanalysis datasets are relatively weak to affect the main results as mentioned above.

### 3.4 Tracers

According to the ventilation theory, if the wind in the north boundary of NWIO was eastward, and the ocean density field had ventilation structure, then the flow in CR was southwestward, and the waters in CR moved from northeast side. Here, the SODA reanalysis supplies compact datasets for passive tracers; therefore, we set some passive tracers along the CR and backtrack their trajectories using the Lagrangian description (Section 2.5), and the results are shown in Fig. 6. In order to better describe the trajectories, we separate the CR to three latitude bands as 8-9.8°N, 5-8°N and 2.3-5°N (equatorial band). For the ASHSW, we set the tracers at a depth of 100 m, and the trajectories reveal different pathways on the CR. In the latitude band of 8-9.8°N (Fig. 6a), the trajectories emphasis the north branch of East African Coastal Current during summer monsoon (Schott and McCreary Jr., 2001; Schott et al., 2009), meanwhile, the water at the north station of CR comes from the northeast side, and one station water shows the cross-equatorial current around 53°E (east of Southern Gyre; Schott et al., 2009). While for latitudes from 5-8°N (Fig. 6c), the water mainly originates from the northeast side, and the trajectories resemble those of the ventilation theory (Luyten et al., 1983; Qiu and Huang, 1995). The results show water coming mainly from the Arabian Basin, and the southwestward flow bring the ASHSW onto the CR. On some occasions, the 100 m depth waters in this latitude section are from the equator, and the pathways show the north branch of East African Coastal Current and the cross-equatorial current around 66°E respectively. Similarly, the pathways starting east of the Horn of Africa are probably due to the off-coast current north of Great Whirl or meso-scale eddies (Chelton et al., 2011; Wang et al., 2019). In the equatorial band (Fig. 6e), the near-equator tracers come from the west side, which is consistent with the north branch of East African Coastal Current. Meanwhile, for the relatively north-side tracers in equatorial band, the trajectories backtrack to east side, which is probably following a westward mean flow, meso-scale eddy or WPPW.

For the RSW in the intermediate-depth layer at 700 m, the trajectories show the zonal movement in the 8-9.8°N (Fig. 6b). Two trajectories move from the west side, which coincident with the potential vorticity explanation (Beal et al., 2000), that the

RSW moves southward along the coast with the help of winter monsoon, and then leave the coast and shift to middle ocean via zonal movement. Meanwhile, there are pathways directly from northwest, and these trajectories support 700 m waters are probably directly from east of the Horn of Africa (or Gulf of Aden) without southward movements along the coast (Shapiro and Meschanov, 1991). In the 5-8°N band (Fig. 6d), the mainly eastward zonal movements agree with Beal et al. (2000), meanwhile, some westward trajectories resemble the prediction of ventilation theory (Luyten et al., 1983; Qiu and Huang, 1995). At last, in the equatorial band (Fig. 6f), most trajectories show eastward zonal movement. Other two trajectories follow westward zonal movement, and one extra trajectory moves from northwest with circular track. Hence, these trajectories display three kinds of pathways.

## 4 Discussion

The CTD and XCTD data are useful in reconstructing the three-dimensional oceanic data. Based on theoretical model, three-dimensional ocean interior can be derived using surface information (SSH and sea surface density), where the reliable mean density field is required (Lapeyre and Klein, 2006; Wang et al., 2013; Liu et al., 2014; Yan et al., 2020). In other words, for the operational purpose, the meso-scale eddy coherent structure could be built on known background density field. The Argo-only data are not sufficient to describe the meso-scale eddy in the NWIO, and they could not supply sufficient background density. In the present study, the maximum distance between Argo profiles is 500 km along the CR; however, after adding the shipboard station data, the maximum distance decreases to 100 km, which falls into the eddy-permitting scale. Sufficient sampling produces more reliable vertical structures of temperature, salinity and density.

The most remarkable signal in the upper ocean is the southward extent of ASHSW, where the counterpart in the climatology data exists but is weak in the horizontal extent. It is surprising that the HYCOM reanalysis captures the phenomenon well, while SODA shows some disadvantages. Although both SODA and HYCOM assimilate the Argo data into Oceanic General Circulation Models (OGCMs), the assimilation methods of SODA and HYCOM are considerably different. SODA adopts optimal interpolation (Carton et al., 2018), while HYCOM uses a 3D varational scheme. One advantage of a 3D variational scheme versus optimal interpolation is the conservation of dynamical constraints (Zhu et al., 2006; Yin et al., 2012; Edwards et al., 2015). Therefore, HYCOM probably describes better the wind-driven circulations, monsoon-induced coastal current and meso-scale eddy movement, which are all related to the southward extent of ASHSW. In the comparative analysis, the state-of-the-art reanalysis is still insufficient to provide good current data. Although similar sea surface dynamic heights are taken into account, the incorrect density field leads to a false vertical structure. It is also noted that the bias is probably further amplified in OGCM and leads to potential unrealistic simulations if these reanalysis data are used in the model for initialization and boundary forcing.

Present study uses SODA reanalysis to investigate the origins of water particles over the CR. However, the corresponding results need further validation. For instance, present study reveals both the meso-scale eddy and WPPW are misinterpreted by SODA, therefore, the waters trapped in meso-scale eddy and WPPW probably move in wrong ways. Meanwhile, trajectories

from different oceanic reanalysis are probably different, regarding the south extents of ASHSW are not same in SODA and HYCOM.

From the theoretical viewpoint, the phase speed of first-, second- and third-mode baroclinic Rossby wave at 6°N in Indian Ocean is roughly 0.6, 0.2 and 0.1 m/s, respectively (Subrahmanyam et al., 2001). The phase speed of WPPW match well with that of the second-mode baroclinic Rossby wave. For the generation mechanics, Subrahmanyam et al. (2001) argued that this kind Rossby wave was probably radiated from coastal trapped Kelvin wave at south-west coast of India. Meanwhile, this kind Rossby wave can bring wave energy from south-west coast of India to the Somali coast, and feed the Somali Current and Somali eddies. The present study displays the vertical structure of this kind Rossby wave, however, the dynamics of WPPW (and Rossby wave) and its association with Somali Current and Somali eddies call for further study.

Generally, the hydrothermal plume in CR uplifts from sea bottom to water depth 2500 m (Murton et al., 2006; Wang et al., 2017). Present paper restricts the sectional study to the upper 1050 m (Fig. 4-5). Within this depth, the water is easily affected by surface forcing. However, on the basin-scale wind-driven circulation, the surface wind forcing affects deeper ocean through quasi-geostrophic instability (Rhines and Young, 1982) and meridional overturning circulation. Meanwhile, the upper 1050 m dynamics over CR is related to the cross-ridge water transport in upper ocean, which influences the deep ocean circulation through pressure adjustment. Therefore, the results provide potential use in the future study of hydrothermal plume.

## 5 Conclusions

This paper reports a onetime hydrographic survey on the CR in the NWIO, where the latitudes cover the equatorial (2.3-5°N) and tropical (5-9.6°N) bands. The station CTD/XCTD sampling plus the Argo floats build the sectional structures of temperature and salinity as well as density. The striking feature is the southern extent of ASHSW from northwest of the CR in the upper ocean. Meanwhile, the temperature and density fields display clear ventilation structures. In the intermediate depth, the observations also capture the RSW at a depth near 700 m.

Furthermore, we compute the absolute geostrophic current based on the density profiles and sea surface height. The vertical structure of the cross-track current reveals strong signals of meso-scale eddies in the upper ocean and relatively weak northeastward current in the intermediate depth. We also identify a strong westward-propagating planetary wave at a latitude of 6°N. The longitude and latitude lengths are 7.5° and 1.3° respectively. The corresponding phase speed is 0.2 m/s, and the vertically affected depth is roughly 200 m.

We further evaluate the state-of-the-art reanalysis with the present observations. As a result, because the Argo profiles and satellite SSH are assimilated into the reanalysis datasets, HYCOM and SODA show relatively good qualities for temperature, salinity and density. However, the reanalysis cross-track currents show large discrepancies compared with the absolute geostrophic current. Most importantly, HYCOM and SODA misinterpret some meso-scale eddies in the current field. Over the NWIO, the meso-scale eddies are relatively important but cannot be well described by the Argo-only data source. The present analysis shows more data source for potential data assimilation experiment. The present situation of insufficient sampling prompts more research activity in the NWIO.

To explore the pathways of ASHSW and RSW during the expedition time, we set tracers in SODA dataset at depths of 100 and 700 m, and backtrack their trajectories via three dimensional Lagrangian description. Overall, for the 100 m depth waters, the results reveal the pathways related to the north branch of East African Coastal Current and the flow from northeast side (or Arabian Basin), while for the 700 m depth waters, the trajectories mainly follow the zonal direction from either west and east

5    sides. The results give direct-viewing descriptions and call for further dynamical investigations.

*Competing interests.*   The authors declare that they have no conflict of interest.

*Acknowledgements.*   This work is supported by the China Ocean Mineral Resources Research and Development Association Project (Nos. DY135-S2-1-07 and DY135-E2-1-01) and the National Natural Science Foundation of China (Nos. 41730535 and 41621064). OSTIA SST data were produced by the Meteorological Office of the United Kingdom (http://marine.copernicus.eu/). We are thankful to editor and two

10    anonymous reviewers for their constructive suggestions and comments. WOA data were available in https://www.nodc.noaa.gov/. Daily Argo float data were downloaded from http://www.argodatamgt.org. CCMP wind were provided by Remote Sensing Systems (http://www.remss.com, version 2.0), AVISO SSH data were produced and distributed by the Copernicus Marine and Environment Monitoring Service (CMEMS) (http://www.marine.copernicus.eu), SODA were obtained from http://www.atmos.umd.edu/ ocean/, and HYCOM were downloaded from https://hycom.org/.

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

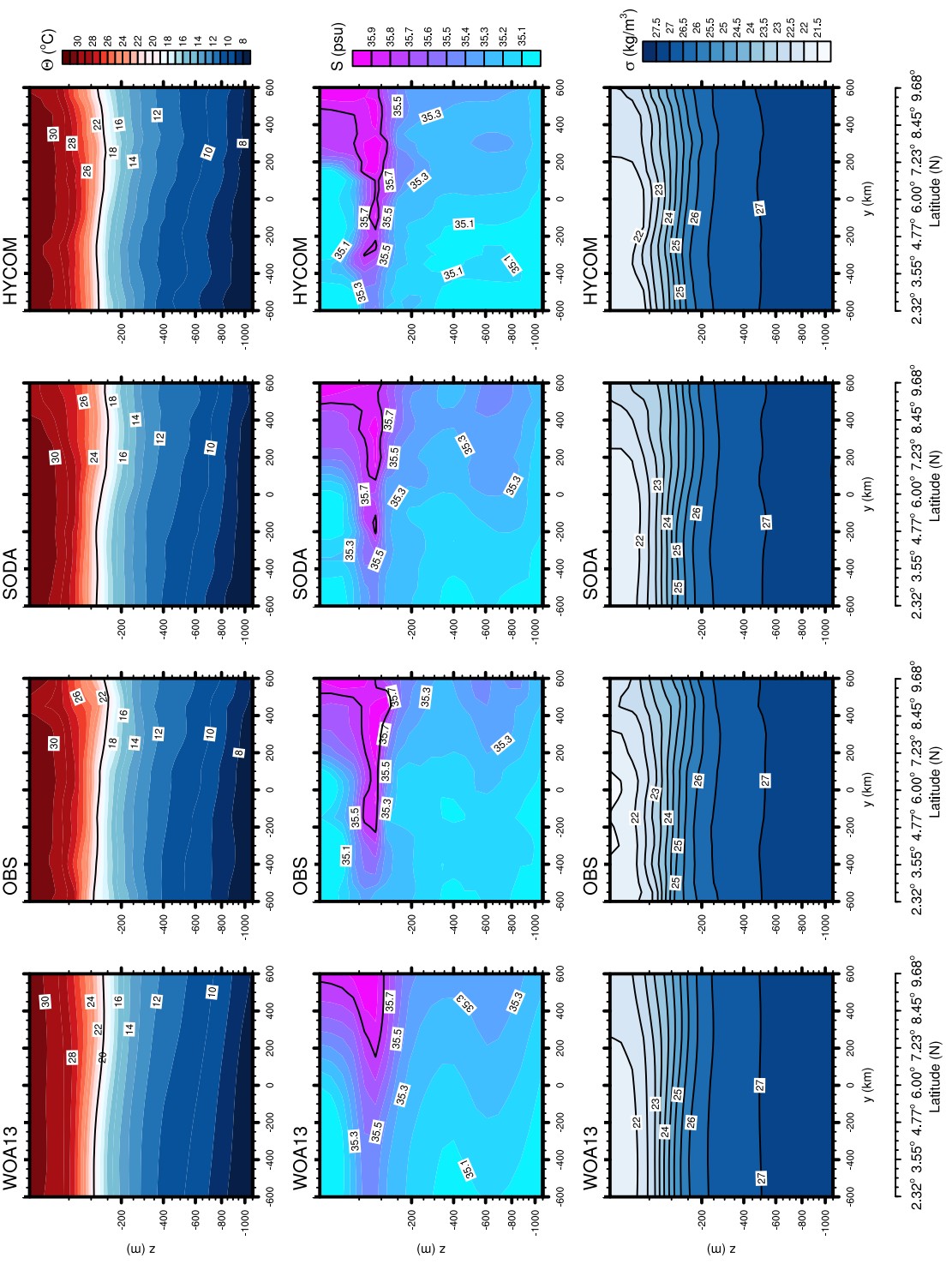

**Figure 4.** Sectional profiles of potential temperature (upper panels), salinity (middle panels) and potential density (lower panels). The data sources cover the WOA climatology (WOA13, version 2.0/A5B2, climatological monthly mean of May), the present observations, and two reanalysis datasets, SODA and HYCOM (monthly mean of May 2012). The isothermal lines of 20°C are highlighted in the potential temperatures, and the iso-salinity lines of 35.8 psu are highlighted in the salinity fields.

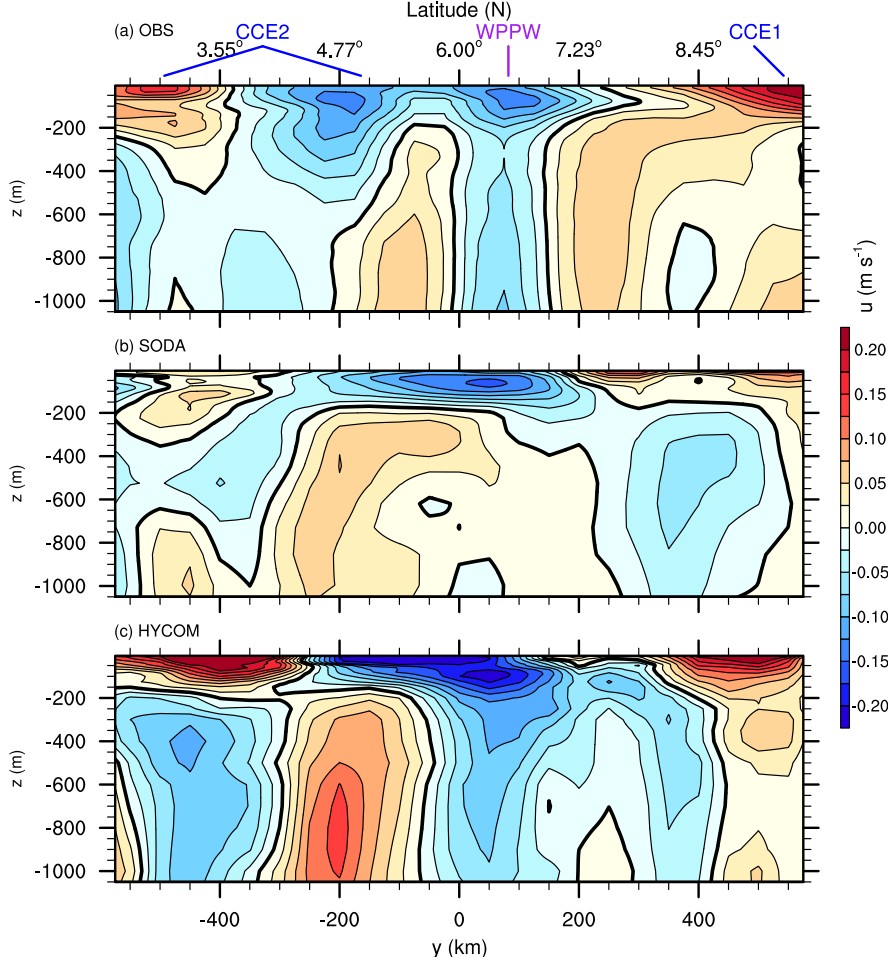

**Figure 5.** Sectional cross-track current: (a) absolute geostrophic current; (b and c) currents in SODA and HYCOM reanalysis, respectively. Northeastward current is positive. Thick black lines are the zero contours.

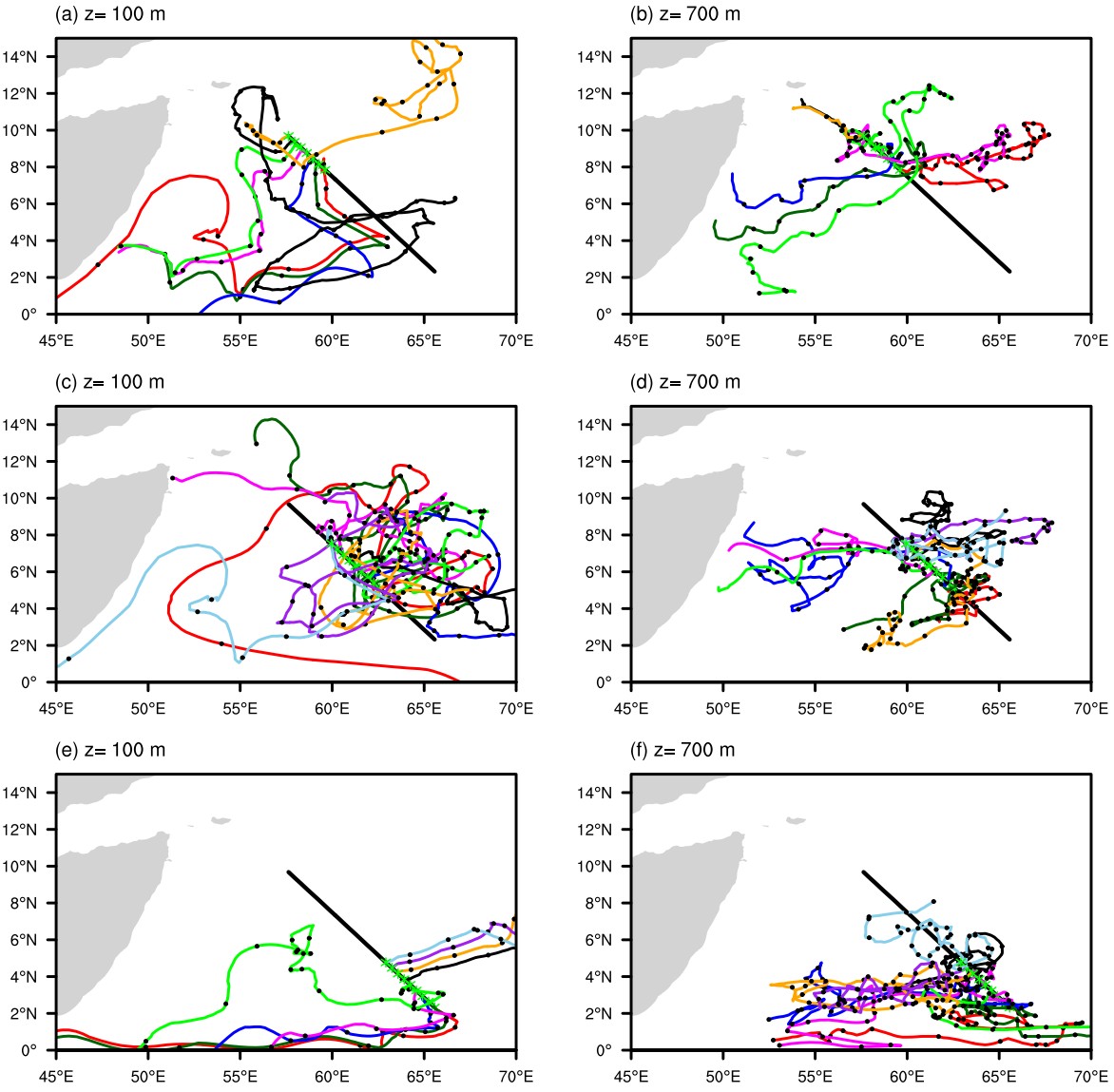

**Figure 6.** Passive tracers using SODA reanalysis. The Carlsberg Ridge is presented by black line. The tracers at May 15, 2012 are shown as green asterisks (ending points). The tracers are backtracked to January 1, 2010. The time interval is one month, as denoted by the black dots. (a-b) 8-9.8°N; (c-d) 5-8°N; (e-f) 2-5°N.