# Peer review of "Hydrographic survey over the Carlsberg Ridge in May 2012"

_Ocean Science, 2019_

## Referee Comment (RC1) · Anonymous Referee #1 · 9 Sep 2019

Based on CTD/XCTD obtained in May 2012 and Argo profiles along the Carlsberg Ridge, this study discussed water masses over the specific section, showed a ventilation structure, and calculated the absolute geostrophic currents. This study is interesting. However, I have the following concerns.

One of the main weaknesses is that this paper does not have a clear scientific theme and draws some quick conclusions. The authors first showed the observed temperature, salinity, and density. Then, they calculated the geostrophic currents and compared the results with SODA and HYCOM. Finally, they set some passive tracers at 100m and 700m and tried to reveal pathways of the masses. It is difficult for the readers to understand what the paper focused on, just like we cannot obtain enough information in science from the title "Hydrographic survey over the Carlsberg Ridge in May 2012".

[Figure]

Because of no in-depth analysis, the paper looks like a data analysis report, and the conclusions are not very convinced.

I thus suggest the authors focus on the water masses in the NWIO. The authors may want to start their analysis based on the CTD/XCTD data. Then, the obtained conclusions are expected to be verified using more data (e.g. more Argo profiles) in this region and more conclusions are expected. Finally, the analysis in section 3.4 is expected to be more solid to reveal the pathway of the water masses. The authors may want to delete the contents in section 3.3.

Some minor comments: 1. Could you add the latitude (or longitude) in the x-axis of Fig. 3? 2. "Wind and SST have a close relationship." The wind pattern may be not a key factor for the SST distribution in the oceanic interior.

---

## Referee Comment (RC2) · Anonymous Referee #2 · 10 Sep 2019

This manuscript introduces hydrographic observations collected over the Carlsberg Ridge in the northwest Indian Ocean. The authors have created a hydrographic dataset that comprises observations collected by a ship and by Argo floats. They identify key water masses, conduct particle tracking experiments, and use the observations to evaluate two re-analysis products. With a few exceptions, the standard of written English is acceptable and the quality of the figures is, on the whole, satisfactory. I think that this manuscript will require substantial revision prior to publication.

Point 1. My primary criticism of this manuscript is that it lacks a clear message; I am unsure of what it is that the authors want me to remember as being important or new. The authors correctly point out that the northwest Indian Ocean is not well sampled, and so any new observations from this region are of value. However, beyond

simply presenting these new observations, the authors, in my view, do not sufficiently demonstrate what we can learn from them. The discussion section, which is where the value of the new observations should be made explicit, largely restates points already made in the results section. It does not cite a single piece of literature. I think that the discussion section needs to be substantially revised: it should explain the value of the observations in the context of relevant literature and, ideally, it should set out a clear argument.

Point 2. The methods outlined in section 2 need to be better explained. The authors note the depth-mean offsets between temperature and salinity observations collected by the ship's CTD and the expendable CTDs (xCTD), and between the xCTDs and an Argo float. Firstly, the implication is that these offsets used to calibrate the xCTD observations – but this should be stated explicitly. Secondly, it is not clear to me whether: (1) xCTD observations are being compared to both ship and Argo observations; or (2) whether ship observations are being compared to xCTD observations, which are then being compared to Argo observations. If the former, which of the two sets of offsets are the authors using for the calibration? If the latter, are the authors calibrating observations from all Argo floats using an offset calculated from just one Argo float? Furthermore, do publicly available observations from the Argo programme need to be calibrated? Are they already calibrated when they are made available for download?

Point 3. The authors point out several times that adding the Argo observations to the ship CTD and XCTD observations enables them to examine mesoscale processes. This may well be the case, but I think that they need to carefully consider the temporal and spatial scales of mesoscale activity in the Arabian Sea. For instance, they include Argo observations from up to 200 km from their section over the Carlsberg Ridge: is this distance less than the Rossby radius at this latitude? Furthermore, Table 1 indicates that the observations were collected over a period of one month. Are the authors confident that these observations may be presented in one section (Figure 4) as if they were a synoptic snapshot?
Line-by-line comments. Page 1, line 14. I am not sure what the authors mean by "renewed" in this context

Page 1, lines 15 – 17. I am not really sure what this sentence means

Page 1, paragraph beginning line 21. This paragraph is not relevant

Page 2, paragraph beginning line 13. This paragraph outlines the reasons for studying the hydrography of the Carlsberg Ridge region, but none of the points raised is revisited later in the paper. The paper would be much improved if, when discussing the results, the authors revisited some of these points – for instance, saying how these new observations help to determine the movements of sporadic hydrothermal activity.

Page 2, line 17. I do not agree that the results presented "shed new light into the basic energy theory of ocean circulation"

Page 2, line 33. It would be nice to have the distance between the CTD and the xCTD station given in km.

Page 3, line 6. Again, it would be nice to have the distance between the Argo and the xCTD station given in km.

Page 4, Figure 1. I think that panel (a) is too small to be of much use: I can't really see the detail because the symbols are too close together. Panel (a) might work better as a separate figure. Similarly, I cannot distinguish individual vorticity contours in panel (b). Contours are not labelled, and the contour interval is not given. The vorticity field should be presented using filled contours, similar to the ADT field in panel (d).

Page 5, line 28. I assume that the data extracted from the re-analysis products are along the same section as the observations, but this should be said explicitly.

Page 6, line 8 (equations). The symbol w is not defined. I assume it is vertical velocity? The authors should state whether they are performing their particle tracking experiments using 2D or 3D velocity.

Page 6, line 10. The authors give the start and end time of the particle tracking experiments here, and not in the caption of Figure 6.

Page 6, line 19. I do not understand what the authors mean when they say that the "wind stress curl highlights the strong seasonal variations". Also, wind stress curl is not shown in a figure.

Page 6, line 27. The statement that warm-core eddies "seem to release footprints in the wind stress curl" does not make sense. I would have thought that the wind influences the eddies and not the other way around. Furthermore, when talking about wind stress curl, the authors refer to Figure 1 (b), but the figure caption says that this panel shows wind velocity and vorticity, not wind stress curl.

Page 6, 29. It is not clear why the westward current is "remarkable". Has this feature not been observed before? Is it significant? Do the authors believe that it cannot be explained by their observations, or by existing theory?

Page 7, line 7. Strictly speaking, the boundary between the tropics and the subtropics is around 23.5°N, so all of the observations being considered here are from the tropics. Consequently, the use of the word subtropical is misleading. Subsequent uses of these terms should also be revised.

Page 7, line 7. I do not think the authors are justified in grouping PGW and RSW as RSPGIW. In Figure 2, the water masses are observed to be separate, and the authors have acknowledged the different densities of the two water masses.

Page 7, line 18. The observations from the World Ocean Atlas presented in Figure 3 are from a climatology and should not, therefore, be described as a snapshot.

Page 7, line 19. The authors have not marked north and south on Figure 3, so it is difficult for the reader to make sense of statements such as "the thermocline... deepens northward".

Page 7, line 22. I do not really agree with the authors' point about ventilation. The

outcropping of isotherms visible in Figure 3 is from within the mixed layer – it is not clear that "subsurface" water is then able to take part in air-sea interaction.

Page 7, line 28. The authors do not explain why their observations show that saline intermediate waters are PGW and RSW – nor do they explain why this is not clear in the climatology.

Page 7, line 31. Why have the authors chosen the 22 kg m-3 isopycnal?

Page 7, line 33. It is hard to compare Figure 1b and Figure 3, because Figure 1b uses degrees and Figure 3 uses km.

Page 7, line 333. I do not understand what is meant by projecting the outcrop point in the wind vector field.

Page 8, Figure 2. I am not sure why the mean temperature-salinity curves have been plotted. They are not mentioned in the text and they distract from the new observations.

Page 9, line 1. I don't think that the thermocline can be described as descending "sharply".

Page 9, line 5. I don't understand what is "remarkable" about the southward extension of salty water.

Page 9, line 15. Again, I don't understand why the velocity field in the upper ocean is "remarkable".

Page 9, line 19. I think the authors need to label features of interest – eg CCE2 – on Figure 4. As it stands, it is quite hard to see what the authors want the reader to look at.

Page 9, line 22. The authors need to discuss, somewhere in the paper, the significance of the westward-propagating disturbance. At the end of the paper, I have no better idea of what is it and why it might be significant than I had at the start.

Page 9, line 26. When comparing the observationally derived geostrophic current and the current fields from the re-analysis products, have the authors considered that the current fields in the re-analysis products might contain ageostrophic components, eg Ekman flow?

Page 10, first paragraph. I do not understand the argument that this paragraph is trying to make. There are several ideas that are not fully explored and which are insufficiently referenced. The authors seem to be contrasting "customary ventilation theory" and its corresponding meridional flow with "potential vorticity analysis" and its corresponding zonal flow. This strikes me as being a massive oversimplification; at the least, it requires a much more detailed explanation. Much of this material – as well as the extra explanation I would like to see added – probably belongs in the introduction.

Page 10, line 1. In what way are explanations of ASHSW and RSW pathways "ambiguous"? This statement needs to be referenced, evidenced and more fully explained, either here or in the introduction.

Page 10, line 5. This sentence is poorly expressed.

Page 10, second paragraph. The authors should explain clearly why they break up the particle tracking results into latitudinal bands – is this because they suspect different processes/currents are causing differences in circulation between these bands? Also, this paragraph should make some attempt to elucidate these processes and to explain what's new and important about these results. At present, the text just explains what the reader can already see in the figure.

Page 10, line 14. I do not agree that this looks like flow in the summer Somali Current.

Page 10, third paragraph. Again, some discussion of processes is needed here.

Page 10, line 25. Are there papers or technical reports available that explain methodological differences between the Hycom and Soda re-analyses? Would more thorough research negate the need for speculation?

Page 10, line 27. It is not clear to me what is meant by the "dynamic core" of the ocean general circulation model.

Page 10, line 33. I dislike the description of geostrophic current as an "alternative result for the ocean current". Geostrophic flow is an important part of ocean circulation and is perfectly valid in its own right.

Page 11, conclusions section. The particle tracking results are not mentioned in the conclusion.

Page 11, line 2. The authors have not discussed baroclinic modes in the results section, so it does not make any sense to the reader when the concept is introduced in the discussion section. Furthermore, are the authors certain that baroclinic mode is an appropriate concept in this instance?

Page 11, line 11. It is not correct to say that you integrate density to get the geostrophic current.

Page 11, line 20. I do not think that the authors have shown how "the present analysis shows potential data applications for the future".

Page 14, Figure 3. Using km as the horizontal co-ordinate is unhelpful, given that in the text what is actually interesting is the location in degrees north. The authors should also mark on features of interest discussed in the text, such as the eddies.

Page 16, Figure 5. Given that the westward-propagating disturbance is so little discussed, I do not think that this figure adds anything to the paper.
* * *

---

## Author Comment (AC1) · 31 Jan 2020

**Response to reviewer # 1**

Based on CTD/XCTD obtained in May 2012 and Argo profiles along the Carlsberg Ridge, this study discussed water masses over the specific section, showed a ventilation structure, and calculated the absolute geostrophic currents. This study is interesting. However, I have the following concerns.

One of the main weaknesses is that this paper does not have a clear scientific theme and draws some quick conclusions. The authors first showed the observed temperature, salinity, and density. Then, they calculated the geostrophic currents and compared the results with SODA and HYCOM. Finally, they set some passive tracers at 100m and

700m and tried to reveal pathways of the masses. It is difficult for the readers to understand what the paper focused on, just like we cannot obtain enough information in science from the title "Hydrographic survey over the Carlsberg Ridge in May 2012" Because of no in-depth analysis, the paper looks like a data analysis report, and the conclusions are not very convinced.

I thus suggest the authors focus on the water masses in the NWIO. The authors may want to start their analysis based on the CTD/XCTD data. Then, the obtained conclusions are expected to be verified using more data (e.g. more Argo profiles) in this region and more conclusions are expected. Finally, the analysis in section 3.4 is expected to be more solid to reveal the pathway of the water masses. The authors may want to delete the contents in section 3.3.

Response: We agree with reviewer's comments and suggestions. We revised the manuscript by

(1) We added the background on meso-scale eddy and west-propagating disturbance in Introduction Section. According to Maximenko et al. (2005), the vertical structure of west-propagating disturbance has not been observed.

(2) We moved the background of Red Sea Water from Results Section into Introduction Section.

(3) We added discussions on the main results of Lagrangian tracers (Section 3.4).

Therefore the main results include the snapshot of water masses, and the vertical structures of mesoscale eddy and west-propagating disturbance.

Reference

Maximenko, N. A., Bang, B., and Sasaki, H.: Observational evidence of alternating zonal jets in the world ocean, Geophysical Research Letters, 32, L12607, 2005.

Some minor comments:

1. Could you add the latitude (or longitude) in the $x$-axis of Fig. 3?

Response: Corrected.

2. "Wind and SST have a close relationship." The wind pattern may be not a key factor for the SST distribution in the oceanic interior.

Response: [Page 7, line 4-6 in change-track-mode revised manuscript] We deleted the statement "Wind and SST have a close relationship".

---

## Author Comment (AC2) · 31 Jan 2020

**Response to reviewer # 2**

This manuscript introduces hydrographic observations collected over the Carlsberg Ridge in the northwest Indian Ocean. The authors have created a hydrographic dataset that comprises observations collected by a ship and by Argo floats. They identify key water masses, conduct particle tracking experiments, and use the observations to evaluate two re-analysis products. With a few exceptions, the standard of written English is acceptable and the quality of the figures is, on the whole, satisfactory. I think that this manuscript will require substantial revision prior to publication.

Response: We are thankful to review's encouragement and constructive suggestions.

[Figure]

We made major revision on the early version manuscript.

Point 1. My primary criticism of this manuscript is that it lacks a clear message; I am unsure of what it is that the authors want me to remember as being important or new. The authors correctly point out that the northwest Indian Ocean is not well sampled, and so any new observations from this region are of value. However, beyond simply presenting these new observations, the authors, in my view, do not sufficiently demonstrate what we can learn from them. The discussion section, which is where the value of the new observations should be made explicit, largely restates points already made in the results section. It does not cite a single piece of literature. I think that the discussion section needs to be substantially revised: it should explain the value of the observations in the context of relevant literature and, ideally, it should set out a clear argument.

Response: The main motivation is to understand the ocean environment over Carlsberg Ridge (CR). The novelty lies on the extra information brought by in-situ CTD & XCTD data. The sectional snapshot therefore gives the vertical structures of temperature, salinity, density and geostrophic current. In the revised manuscript, we emphasized three valuable results: (1) The snapshot of water masses, (2) the structure of mesoscale eddy, and (3) the structure of west-propagating disturbance. We revised the Introduction and Discussion sections, accordingly.

Point 2. The methods outlined in section 2 need to be better explained. The authors note the depth-mean offsets between temperature and salinity observations collected by the ship's CTD and the expendable CTDs (xCTD), and between the xCTDs and an Argo float. Firstly, the implication is that these offsets used to calibrate the xCTD observations ? but this should be stated explicitly. Secondly, it is not clear to me whether: (1) xCTD observations are being compared to both ship and Argo observations; or (2) whether ship observations are being compared to xCTD observations, which are then being compared to Argo observations. If the former, which of the two sets of offsets are the authors using for the calibration? If the latter, are the authors calibrating

observations from all Argo floats using an offset calculated from just one Argo float? Furthermore, do publicly available observations from the Argo programme need to be calibrated? Are they already calibrated when they are made available for download?
Response: [Page 5, line 5 in change-track-mode revised manuscript] Good suggestions. In section 2.1, we just clarify the differences among CTD, xCTD and Argo float. We did not calibrated the xCTD and Argo float before further low-pass filter processing. To avoid misunderstanding, we add a sentence as
"We later use objective analysis method and low-pass filter to reduce the differences among CTD, XCTD and Argo."

Point 3. The authors point out several times that adding the Argo observations to the ship CTD and XCTD observations enables them to examine mesoscale processes. This may well be the case, but I think that they need to carefully consider the temporal and spatial scales of mesoscale activity in the Arabian Sea. For instance, they include Argo observations from up to 200 km from their section over the Carlsberg Ridge: is this distance less than the Rossby radius at this latitude? Furthermore, Table 1 indicates that the observations were collected over a period of one month. Are the authors confident that these observations may be presented in one section (Figure 4) as if they were a synoptic snapshot?
Response: In the monthly mean sea surface height (Fig. 2c in revised manuscript), two dynamics include meso-scale eddy and west-propagating disturbance are identified, we therefore confirm the two dynamics are beyond the synoptic snapshot.

Line-by-line comments.
Page 1, line 14. I am not sure what the authors mean by "renewed" in this context.
Response: [Page 1, line 14 in change-track-mode revised manuscript] We deleted the "renewed".

Page 1, lines 15–17. I am not really sure what this sentence means.
Response: [Page 1, line 15-16, in change-track-mode revised manuscript]
"Moreover, the monsoon builds up a meridional current in the NWIO, which changing

the form of the customary zonal current (as in the Pacific and Atlantic Oceans) into the meridional current."
was changed to
"Moreover, the monsoon is so strong to change the pattern of basin-scale circulation. The monsoon builds up a dominant meridional current in the NWIO, which changing the form of the customary zonal current (as in the Pacific and Atlantic Oceans) into the meridional current."

Page 1, paragraph beginning line 21. This paragraph is not relevant
Response: We deleted the paragraph.

Page 2, paragraph beginning line 13. This paragraph outlines the reasons for studying the hydrography of the Carlsberg Ridge region, but none of the points raised is revisited later in the paper. The paper would be much improved if, when discussing the results, the authors revisited some of these points for instance, saying how these new observations help to determine the movements of sporadic hydrothermal activity.
Response: [Page 13, line 21-26, in change-track-mode revised manuscript] We added the response to second reason in the Discussion Section, as
"Present paper restricts the sectional study in the upper 1050 m (Fig. 4-5). Within this depth, the water is easy affected by surface forcing. However, on the basin-scale wind-driven circulation, the surface wind forcing affects deeper ocean through quasi-geostrophic instability (Rhines and Young, 1982) and meridional overturning circulation. Generally, the hydrothermal plume in CR uplifts from sea bottom to water depth 2500 m (Murton et al., 2006; Wang et al., 2017). Because present paper concentrates on the CR as well as the cross-ridge current, the results provide potential use in the future study of hydrothermal plume event."

Page 2, line 17. I do not agree that the results presented "shed new light into the basic energy theory of ocean circulation"
Response: [Page 2, line 30, in change-track-mode revised manuscript] We weaken the statement as: "provides a reference to the basic energy theory of ocean circulation".

Page 2, line 33. It would be nice to have the distance between the CTD and the xCTD station given in km.
Response: [Page 4, line 1, in change-track-mode revised manuscript] We added the distance in km

Page 3, line 6. Again, it would be nice to have the distance between the Argo and the xCTD station given in km.
Response: [Page 4, line 6, in change-track-mode revised manuscript] We added the distance in km

Page 4, Figure 1. I think that panel (a) is too small to be of much use: I can't really see the detail because the symbols are too close together. Panel (a) might work better as a separate figure. Similarly, I cannot distinguish individual vorticity contours in panel (b). Contours are not labelled, and the contour interval is not given. The vorticity field should be presented using filled contours, similar to the ADT field in panel (d).
Response: We revised the Figure 1 according to reviewer's suggestions.

Page 5, line 28. I assume that the data extracted from the re-analysis products are along the same section as the observations, but this should be said explicitly.
Response: [Page 6, line 12, in change-track-mode revised manuscript] Yes. We added a statement in section 2.4, as
"For comparison, we extract the reanalysis datasets along the same section as the observation."

Page 6, line 8 (equations). The symbol w is not defined. I assume it is vertical velocity? The authors should state whether they are performing their particle tracking experiments using 2D or 3D velocity.
Response: [Page 6, line 18-19, in change-track-mode revised manuscript] In section 2.5, we insert two sentences, as
"$w$ is the vertical velocity"
and

"we use the three dimensional velocity $(U, V, w)$ to track the tracers,"

Page 6, line 10. The authors give the start and end time of the particle tracking experiments here, and not in the caption of Figure 6.
Response:[Page 6, line 20, in change-track-mode revised manuscript] Corrected.

Page 6, line 19. I do not understand what the authors mean when they say that the "wind stress curl highlights the strong seasonal variations". Also, wind stress curl is not shown in a figure.
Response: [Page 7, line 2, in change-track-mode revised manuscript] We deleted the sentence.

Page 6, line 27. The statement that warm-core eddies "seem to release footprints in the wind stress curl" does not make sense. I would have thought that the wind influences the eddies and not the other way around. Furthermore, when talking about wind stress curl, the authors refer to Figure 1 (b), but the figure caption says that this panel shows wind velocity and vorticity, not wind stress curl.
Response: [Page 7, line 10-11, in change-track-mode revised manuscript] We deleted the sentence.

Page 6, 29. It is not clear why the westward current is "remarkable". Has this feature not been observed before? Is it significant? Do the authors believe that it cannot be explained by their observations, or by existing theory?
Response: [Page 7, line 12, in change-track-mode revised manuscript] We noted the westward current "remarkable" according to the magnitude and zonal extent. As refer to Maximenko et al. (2005), present observation displays the vertical structure of this westward current. The related theory needs to be further confirmed.
Maximenko, N. A., Bang, B., and Sasaki, H.: Observational evidence of alternating zonal jets in the world ocean, Geophysical Research Letters, 32, L12607, 2005.

Page 7, line 7. Strictly speaking, the boundary between the tropics and the subtropics is around 23.5°N, so all of the observations being considered here are from the tropics.

Consequently, the use of the word subtropical is misleading. Subsequent uses of these terms should also be revised.

Response: We corrected the statement accordingly. We changed the "tropical band" to "equatorial band", and changed the "subtropical band" to "tropical band" in the whole paper.

Page 7, line 7. I do not think the authors are justified in grouping PGW and RSW as RSPGIW. In Figure 2, the water masses are observed to be separate, and the authors have acknowledged the different densities of the two water masses.

Response: [Page 7, line 28-29, in change-track-mode revised manuscript] We deleted the statements related to RSPGIW.

Page 7, line 18. The observations from the World Ocean Atlas presented in Figure 3 are from a climatology and should not, therefore, be described as a snapshot.

Response: [Page 7, line 30, in change-track-mode revised manuscript] We corrected the statement.

Page 7, line 19. The authors have not marked north and south on Figure 3, so it is difficult for the reader to make sense of statements such as "the thermocline... deepens northward".

Response: [Page 18, Figure 4, in change-track-mode revised manuscript] We therefore added the latitude in Figure 3 (Figure 4 in revised manuscript).

Page 7, line 22. I do not really agree with the authors's point about ventilation. The outcropping of isotherms visible in Figure 3 is from within the mixed layer? it is not clear that "subsurface" water is then able to take part in air-sea interaction.

Response: [Page 7, line 34, in change-track-mode revised manuscript] We argue that the mixed-layer is approximately well-mixed bulk layer, which can be defined as the layer of temperature within SST minus 0.1 (or 0.5) °C. In Figure 3 (Figure 4 in revised manuscript), the interval of temperature contour is 1.0 °C, therefore, we suppose that the outcropping isothermal line was below the mixed-layer, and argue the "subsurface"

water taken part in air-sea interaction.

Page 7, line 28. The authors do not explain why their observations show that saline intermediate waters are PGW and RSW ? nor do they explain why this is not clear in the climatology.

Response: [Page 10, line 5, in change-track-mode revised manuscript] Basically, we identified the PGW and RSW from T-S diagram. We added the reason in revised manuscript. For explain on the climatology, we added

"In other words, the comparison implies that the year 2002 is an anomaly year on the activities of ASHSW, PGW and RSW."

Page 7, line 31. Why have the authors chosen the 22 kg m-3 isopycnal?

Response: We chosen 22 kg m-3 because it was the first near-surface isopycnal in climatology, meanwhile, and it characterized the differences among four datasets.

Page 7, line 33. It is hard to compare Figure 1b and Figure 3, because Figure 1b uses degrees and Figure 3 uses km.

Response: [Page 18, Figure 4, in change-track-mode revised manuscript] We added the latitude in the Figure.

Page 7, line 33. I do not understand what is meant by projecting the outcrop point in the wind vector field.

Response: [Page 10, line 9, in change-track-mode revised manuscript] We deleted "when we project the outcrop point in the wind vector field".

Page 8, Figure 2. I am not sure why the mean temperature-salinity curves have been plotted. They are not mentioned in the text and they distract from the new observations.

Response: [Page 9, Figure 3, in change-track-mode revised manuscript] We moved out the mean temperature-salinity curves.

Page 9, line 1. I don't think that the thermocline can be described as descending "sharply".

Response: [Page 10, line 14, in change-track-mode revised manuscript] We changed "sharply" to "steeply".

Page 9, line 5. I don't understand what is "remarkable" about the southward extension of salty water.
Response: [Page 10, line 18, in change-track-mode revised manuscript] "the remarkable southward extension of salty water in the upper ocean"
is modified to
"the southward extension of ASHSW".

Page 9, line 15. Again, I don't understand why the velocity field in the upper ocean is "remarkable".
Response: [Page 10, line 28, in change-track-mode revised manuscript] We changed "remarkable" to "relatively strong".

Page 9, line 19. I think the authors need to label features of interest? eg CCE2 ? on Figure 4. As it stands, it is quite hard to see what the authors want the reader to look at.
Response: [Page 19 Figure 5, in change-track-mode revised manuscript] We added the position of CCE1, CCE2 and WPD in the Figure.

Page 9, line 22. The authors need to discuss, somewhere in the paper, the significance of the westward-propagating disturbance. At the end of the paper, I have no better idea of what is it and why it might be significant than I had at the start.
Response: [Page 2 line 22-24, in change-track-mode revised manuscript] In the introduction section, we added:
"The planetary waves at-least include Rossby wave, Kelvin wave and west-propagating disturbance (Rhines, 1975; McCreary, 1985). Specifically, the vertical structure of the west-propagating disturbance needs further investigation in NWIO (Maximenko et al., 2005)."

Page 9, line 26. When comparing the observationally derived geostrophic current and

the current fields from the re-analysis products, have the authors considered that the current fields in the re-analysis products might contain ageostrophic components, eg Ekman flow?

Response: [Page 11 line 17-22, in change-track-mode revised manuscript] In section 3.3, we added a paragraph on discussion the Ekman flow:

"Part of the difference between observation-based absolute geostrophic current and re-analysis current is due to the near-surface Ekman current. The climatological monthly mean mixed-layer depth in May is roughly 20 m at station 9.5°N and 59.5°E (on CR; Liu et al., 2018). Besides, the mean surface Ekman speed, which is approximately the difference between surface geostrophic current and in-situ surface current (from sur-face drifter), is within 0.1 m/s in northern IO (Saj, 2017). Therefore, the near-surface Ekman currents in reanalysis datasets are relatively weak to affect the main results as mentioned above."

Page 10, first paragraph. I do not understand the argument that this paragraph is trying to make. There are several ideas that are not fully explored and which are insufficiently referenced. The authors seem to be contrasting "customary ventilation theory" and its corresponding meridional flow with "potential vorticity analysis" and its corresponding zonal flow. This strikes me as being a massive oversimplification; at the least, it re-quires a much more detailed explanation. Much of this material " as well as the extra explanation I would like to see added " probably belongs in the introduction.

Page 10, line 1. In what way are explanations of ASHSW and RSW pathways "am-biguous". This statement needs to be referenced, evidenced and more fully explained, either here or in the introduction.

Response: [Page 2 line 10-16, in change-track-mode revised manuscript] We modified the explanation, as

"For instance, three water masses were defined in NWIO as Arabian Sea High-Salinity Water (ASHSW), Persian Gulf Water (PGW) and Red Sea Water (RSW). Regarding the pathways of these water masses, the mechanics are not clear. RSW is formed near the northern side of the NWIO; therefore, according to the customary ocean ventilation

theory, RSW sinks and moves southward along the isopycnal layer from the generation zone following the wind-driven current (Luyten et al., 1983). However, the feasibility of the ocean ventilation theory is still unknown for the northern IO, whose meridional extent is limited compared with the other two basins. In contrast, in situ potential vorticity analysis on RSW reveals that the flows generally follow the zonal direction (Beal et al., 2000)."
We moved the paragraph into the Introduction Section.

Page 10, line 5. This sentence is poorly expressed.
Response: [Page 2 line 14-15, in change-track-mode revised manuscript] We changed "However, the feasibility of the ocean ventilation theory is still under debate, especially for the northern IO, whose meridional extent is limited compared with the other two basins. "
into
"However, the feasibility of the ocean ventilation theory is still unknown for the northern IO, whose meridional extent is limited compared with the other two basins."
We moved the paragraph into the Introduction Section.

Page 10, second paragraph. The authors should explain clearly why they break up the particle tracking results into latitudinal bands? is this because they suspect different processes/currents are causing differences in circulation between these bands? Also, this paragraph should make some attempt to elucidate these processes and to explain what's new and important about these results. At present, the text just explains what the reader can already see in the figure.
Page 10, line 14. I do not agree that this looks like flow in the summer Somali Current.
Response: [Page 11, line 35 – Page 12 line 15, in change-track-mode revised manuscript] We revised the paragraph. We added:
"For better describing the trajectories, we separate the CR to three latitude bands as 2.3-5°N (equatorial band), 5-8°N and 8-9.8°N."
"In the equatorial band (Fig. 6e), the near-equator tracers come from the west side,

which is consistent with the north branch of East African Coastal Current during summer monsoon (Schott and McCreary Jr., 2001; Schott et al., 2009)."

"Meanwhile, for the relatively north-side tracers in equatorial band, the trajectories backtrack to east side, which is probably following a westward current or meso-scale eddy."

"For latitudes from 8-9.8°N (Fig. 6a), the trajectories emphasis the north branch of East African Coastal Current, meanwhile, the water at the north station of CR comes from the northeast side, and one station water shows the cross-equatorial current around 53°E (east of Southern Gyre; Schott et al., 2009). "

Accordingly, we deleted:

"For the tropical band (Fig. 6e), the water mainly follows the zonal movement, but the near-equator tracers are from west side and relatively north-side tracers come from east side."

"For latitudes from 8-9.8°N (Fig. 6a), the trajectories look like the flow of the summer Somalia Current (Schott et al., 2009)"

Page 10, third paragraph. Again, some discussion of processes is needed here.
Response: [Page 12, line 16-27, in change-track-mode revised manuscript] We rewritten the paragraph. Original version:

For the RSW in the intermediate-depth layer at 700 m, the trajectories in the tropical band (Fig. 6f) and at latitudes from 5-8°N (Fig. 6d) generally follow the zonal movement (Beal et al., 2000). The tracer movements at latitudes from 8-9.8°N (Fig. 6b) partly agree with those of the ventilation theory, and partly follow the zonal direction (Beal et al., 2000).

New version:

For the RSW in the intermediate-depth layer at 700 m, the trajectories in the equatorial band (Fig. 6f) show the zonal movement. Most of the trajectories move from the west side, which coincident with the potential vorticity explanation (Beal et al, 2000), that the RSW moves southward along the coast with the help of winter monsoon, and then leave the coast and shift to middle ocean via zonal movement. Other two trajectories

come from east side, and one extra trajectory moves from northwest with circular track. Hence, these trajectories display three kinds of pathways. Accordingly, in the 5-8°N band (Fig. 6d), the mainly eastward zonal movements agree with (Beal et al., 2000), meanwhile, some westward trajectories resemble the ventilation theory (Luyten et al., 1983; Qiu and Huang, 1995). At last, in the 8-9.8°N (Fig. 6b), new pathway directly from northwest is emerged, and the trajectories support 700 m waters are probably directly from east of the Horn of Africa (or Gulf of Aden) without southward movements along the coast.

Page 10, line 25. Are there papers or technical reports available that explain methodological differences between the HYCOM and SODA re-analyses? Would more thorough research negate the need for speculation?
Page 10, line 27. It is not clear to me what is meant by the "dynamic core" of the ocean general circulation model.
Response: [Page 13, line 3-14, in change-track-mode revised manuscript] We nearly rewritten the paragraph. We changed
"We speculate that although both SODA and HYCOM assimilate the Argo data into an Oceainc General Circulation Model (OGCM), the methodology of assimilation or the weight between OGCM and in situ observations, is sharply different. We assume the southward extension of ASHSW could be simulated by the dynamic core of OGCM, and the phenomenon was not captured in the Argo-only observation, therefore, HYCOM seems more to approach the dynamic model, and SODA weighted more on the Argo-only observations. Additionally, the finer horizontal resolution of HYCOM likely helps HYCOM involve more physically sound mechanics, such as the downwelling of salty water and wind-driven meridional movement."
to
"Although both SODA and HYCOM assimilate the Argo data into Oceainc General Circulation Models (OGCMs), the assimilation methods of SODA and HYCOM are considerably different. SODA adopts optimum interpolation (Carton et al., 2008), while HYCOM uses 3D variation scheme. One advantage of 3D variation scheme versus

optimum interpolation is the conservation of dynamical constrains (Zhu et al., 2006; Yin et al., 2012; Edwards et al., 2015). Therefore, HYCOM probably describes better on the wind-driven circulations, monsoon-induced coastal current and meso-scale eddy movement, which are all related to the southward extension of ASHSW."

Page 10, line 33. I dislike the description of geostrophic current as an "alternative result for the ocean current". Geostrophic flow is an important part of ocean circulation and is perfectly valid in its own right.
Response: [Page 13, line 15-17, in change-track-mode revised manuscript] The sentences were deleted.

Page 11, conclusions section. The particle tracking results are not mentioned in the conclusion.
Response: [Page 14, line 14-18, in change-track-mode revised manuscript] We added paragraph in conclusion section, as
"To explore the pathways of ASHSW and RSW during the expedition time, we set tracers in SODA dataset at depths of 100 and 700 m, and backtrack their trajectories via three dimensional Lagrangian description. Overall, for the 100 m depth waters, the results reveal the pathways related to the north branch of East African Coastal Current and the flow from northeast side (or Arabian Basin), while for the 700 m depth waters, the trajectories mainly follow the zonal direction from either west and east sides. The results give direct-viewing descriptions and call for further dynamical investigations."

Page 11, line 2. The authors have not discussed baroclinic modes in the results section, so it does not make any sense to the reader when the concept is introduced in the discussion section. Furthermore, are the authors certain that baroclinic mode is an appropriate concept in this instance?
Response: [Page 13, line 18, in change-track-mode revised manuscript] We changed the word "baroclinic" into "vertical".

Page 11, line 11. It is not correct to say that you integrate density to get the geostrophic

current.

Response: [Page 14, line 1, in change-track-mode revised manuscript] We changed "we integrate the density field to obtain the absolute geostrophic current."
into
"we compute the absolute geostrophic current based on the density profiles and sea surface height".

Page 11, line 20. I do not think that the authors have shown how "the present analysis shows potential data applications for the future".

Response: [Page 14, line 9-11, in change-track-mode revised manuscript] We changed "The present analysis shows potential data applications for the future, where the meso-scale eddies are relatively important but cannot be well described by the Argo-only data source."
into
"Over the NWIO, the meso-scale eddies are relatively important but cannot be well described by the Argo-only data source. The present analysis shows more data source for potential data assimilation experiment."

Page 14, Figure 3. Using km as the horizontal co-ordinate is unhelpful, given that in the text what is actually interesting is the location in degrees north. The authors should also mark on features of interest discussed in the text, such as the eddies.

Response: [Page 18, Figure 4, in change-track-mode revised manuscript] We added the latitude in the Figure.

Page 16, Figure 5. Given that the westward-propagating disturbance is so little discussed, I do not think that this figure adds anything to the paper.

Response: We removed this figure.

---

## Author Response (AR2)

**Response to reviewer # 1**

Based on CTD/XCTD obtained in May 2012 and Argo profiles along the Carlsberg Ridge, this study discussed water masses over the specific section, showed a ventilation structure, and calculated the absolute geostrophic currents. This study is interesting. However, I have the following concerns.

One of the main weaknesses is that this paper does not have a clear scientific theme and draws some quick conclusions. The authors first showed the observed temperature, salinity, and density. Then, they calculated the geostrophic currents and compared the results with SODA and HYCOM. Finally, they set some passive tracers at 100m and 700m and tried to reveal pathways of the masses. It is difficult for the readers to understand what the paper focused on, just like we cannot obtain enough information in science from the title "Hydrographic survey over the Carlsberg Ridge in May 2012" Because of no in-depth analysis, the paper looks like a data analysis report, and the conclusions are not very convinced.

I thus suggest the authors focus on the water masses in the NWIO. The authors may want to start their analysis based on the CTD/XCTD data. Then, the obtained conclusions are expected to be verified using more data (e.g. more Argo profiles) in this region and more conclusions are expected. Finally, the analysis in section 3.4 is expected to be more solid to reveal the pathway of the water masses. The authors may want to delete the contents in section 3.3.

Response: We agree with reviewer's comments and suggestions. We revised the manuscript by (1) We added the background on meso-scale eddy and west-propagating disturbance (changed to "west-propagating planetary wave (WPPW)") in Introduction Section, like

"Besides, the planetary waves at-least include Rossby and Kelvin waves (Rhines, 1975; Mc-Creary, 1985). Satellite-retrieved sea surface height are commonly used to detect the planetary waves (Chelton and Schlax, 1996), however, the internal dynamics of planetary wave is not sufficiently addressed. The phase speed of west-propagating planetary wave (WPPW) is used to match the theoretical Rossby wave, nonetheless, the vertical structure of WPPW calls for vertical profiling observation (Subrahmanyam et al., 2001)."

(2) We moved the background of Red Sea Water from Results Section into Introduction Section, as

"For instance, three water masses were defined in NWIO as Arabian Sea High-Salinity Water (ASHSW), Persian Gulf Water (PGW) and Red Sea Water (RSW). Regarding the pathways of these water masses, the mechanics are not clear. RSW is formed near the northern side of the NWIO; therefore, according to the customary ocean ventilation theory, RSW sinks and moves southward along the isopycnal layer from the generation zone following the wind-driven current (Luyten et al., 1983). However, the applicability of the ocean ventilation theory is still unknown for the northern IO, because the surface wind reverses direction under the influence of wind monsoon (Liu et al., 2018). Meanwhile, the limited meridional extent of IO omits the polar-to-subpolar front, which helps form intermediate water in Pacific Ocean and Atlantic Ocean (You, 1998). In contrast, in situ potential vorticity analysis on RSW reveals that the

flows generally follow the zonal direction (Beal et al., 2000). How the water mass moves is worthy further investigation."

(3) We added discussions on the main results of Lagrangian tracers (Section 3.4).

Therefore the main results include the snapshot of water masses, and the vertical structures of mesoscale eddy and west-propagating planetary wave.

Some minor comments: 1. Could you add the latitude (or longitude) in the *x*-axis of Fig. 3? Response: Corrected.

2. "Wind and SST have a close relationship." The wind pattern may be not a key factor for the SST distribution in the oceanic interior.

Response: We deleted the statement "Wind and SST have a close relationship".

**Response to reviewer # 2**

This manuscript introduces hydrographic observations collected over the Carlsberg Ridge in the northwest Indian Ocean. The authors have created a hydrographic dataset that comprises observations collected by a ship and by Argo floats. They identify key water masses, conduct particle tracking experiments, and use the observations to evaluate two re-analysis products. With a few exceptions, the standard of written English is acceptable and the quality of the figures is, on the whole, satisfactory. I think that this manuscript will require substantial revision prior to publication.

Response: We are thankful to review's encouragement and constructive suggestions. We made major revision on the early version manuscript.

Point 1. My primary criticism of this manuscript is that it lacks a clear message; I am unsure of what it is that the authors want me to remember as being important or new. The authors correctly point out that the northwest Indian Ocean is not well sampled, and so any new observations from this region are of value. However, beyond simply presenting these new observations, the authors, in my view, do not sufficiently demonstrate what we can learn from them. The discussion section, which is where the value of the new observations should be made explicit, largely restates points already made in the results section. It does not cite a single piece of literature. I think that the discussion section needs to be substantially revised: it should explain the value of the observations in the context of relevant literature and, ideally, it should set out a clear argument.

Response: The main motivation is to understand the ocean environment over Carlsberg Ridge (CR). The novelty lies on the extra information brought by in-situ CTD & XCTD data. The sectional snapshot therefore gives the vertical structures of temperature, salinity, density and geostrophic current. In the revised manuscript, we emphasized three valuable results: (1) The snapshot of water masses, (2) the structure of mesoscale eddy, and (3) the structure of west-propagating disturbance (changed to "west-propagating planetary wave"). We revised the Introduction and Discussion sections, accordingly.

Point 2. The methods outlined in section 2 need to be better explained. The authors note the depth-mean offsets between temperature and salinity observations collected by the ship's CTD and the expendable CTDs (xCTD), and between the xCTDs and an Argo float. Firstly, the implication is that these offsets used to calibrate the xCTD observations ? but this should be stated explicitly. Secondly, it is not clear to me whether: (1) xCTD observations are being compared to both ship and Argo observations; or (2) whether ship observations are being compared to xCTD observations, which are then being compared to Argo observations. If the former, which of the two sets of off- sets are the authors using for the calibration? If the latter, are the authors calibrating observations from all Argo floats using an offset calculated from just one Argo float? Furthermore, do publicly available observations from the Argo programme need to be calibrated? Are they already calibrated when they are made available for download? Response: [Page 3, line 28–31 in change-track-mode revised manuscript] Thanks for the suggestions. We did not calibrate the instrumental readings. XCTD observations were compared to both ship CTD and Argo observation. However, every XCTD and Argo has individual sensor, we did not impose calibration on these datasets. In the revised manuscript, we stated as:

"Differences among CTD, XCTD and Argo are not negligible. However, because the distances between two stations in comparisons are relatively large, and the biases of XCTD and Argo are different for instruments, we does not perform instrument calibration."

In addition, Argo data are not need further calibrated, the official site already made data quality control.

Point 3. The authors point out several times that adding the Argo observations to the ship CTD and XCTD observations enables them to examine mesoscale processes. This may well be the case, but I think that they need to carefully consider the temporal and spatial scales of mesoscale activity in the Arabian Sea. For instance, they include Argo observations from up to 200 km from their section over the Carlsberg Ridge: is this distance less than the Rossby radius at this latitude? Furthermore, Table 1 indicates that the observations were collected over a period of one month. Are the authors confident that these observations may be presented in one section (Figure 4) as if they were a synoptic snapshot?

Response: In the monthly mean sea surface height (Fig. 2c in revised manuscript), two dynamics include meso-scale eddy and west-propagating planetary wave are identified, we therefore confirm the two dynamics are beyond the synoptic snapshot.

Line-by-line comments.

Page 1, line 14. I am not sure what the authors mean by "renewed" in this context. Response: [Page 1, line 14 in change-track-mode revised manuscript] We deleted the "renewed".

Page 1, lines 15–17. I am not really sure what this sentence means.

Response: [Page 1, line 15-16, in change-track-mode revised manuscript]

"Moreover, the monsoon builds up a meridional current in the NWIO, which changing the form of the customary zonal current (as in the Pacific and Atlantic Oceans) into the meridional current."

was changed to

"Moreover, the monsoon is strong to change the pattern of basin-scale circulation. The monsoon builds up a dominant meridional current in the NWIO, changing the form of the customary zonal current (as in the Pacific and Atlantic Oceans) into a meridional current."

Page 1, paragraph beginning line 21. This paragraph is not relevant Response: We deleted the paragraph.

Page 2, paragraph beginning line 13. This paragraph outlines the reasons for studying the hydrography of the Carlsberg Ridge region, but none of the points raised is revisited later in the paper. The paper would be much improved if, when discussing the results, the authors revisited some of these points for instance, saying how these new observations help to determine the movements of sporadic hydrothermal activity.

Response: [Page 2, line 32–35; and Page 14, line 18–25, in change-track-mode revised manuscript] We revised the motivation of present work at the end of Introduction:

"First, hydrographic survey takes a snapshot on the water mass, and gives an evidence on the activity of water mass. Second, the observation probably captures the vertical structure of meso-scale eddy or planetary wave. Third, the results could be used to evaluate the widely-used oceanic reanalysis. "

Furthermore, we added the discussion involving hydrothermal activity in Discussion Section, as

"Generally, the hydrothermal plume in CR uplifts from sea bottom to water depth 2500 m (Murton et al., 2006, Wang et al., 2017). Present paper restricts the sectional study to the upper 1050 m (Fig. 4-5). Within this depth, the water is easily affected by surface forcing. However, on the basin-scale wind-driven circulation, the surface wind forcing affects deeper ocean through quasi-geostrophic instability (Rhines and Young, 1982) and meridional over-turning circulation. Because present paper concentrates on the CR as well as the cross-ridge current, the upper 1050 m dynamics towards the upper ocean cross-ridge water transport, which is related to the deeper ocean dynamics through pressure adjustment, meanwhile, the activity of meso-scale eddy induces deeper ocean vorticity response (Rhines and Young, 1982), the results provide potential use in the future study of hydrothermal plume event."

Page 2, line 17. I do not agree that the results presented "shed new light into the basic energy theory of ocean circulation"

Response: [Page 3, line 2, in change-track-mode revised manuscript] We deleted the statement..

Page 2, line 33. It would be nice to have the distance between the CTD and the xCTD station given in km.

Response: [Page 3, line 18, in change-track-mode revised manuscript] We added the distance in km

Page 3, line 6. Again, it would be nice to have the distance between the Argo and the xCTD station given in km.

Response: [Page 3, line 24, in change-track-mode revised manuscript] We added the distance in km

Page 4, Figure 1. I think that panel (a) is too small to be of much use: I can't really see the detail because the symbols are too close together. Panel (a) might work better as a separate figure. Similarly, I cannot distinguish individual vorticity contours in panel (b). Contours are not labelled, and the contour interval is not given. The vorticity field should be presented using filled contours, similar to the ADT field in panel (d).

Response: We revised the Figure 1 according to reviewer's suggestions.

Page 5, line 28. I assume that the data extracted from the re-analysis products are along the same section as the observations, but this should be said explicitly.

Response: [Page 6, line 19, in change-track-mode revised manuscript] Yes. We added a statement in section 2.4, as

"For comparison, we extract the reanalysis datasets along the same section as the observations, and the monthly mean fields are used."

Page 6, line 8 (equations). The symbol w is not defined. I assume it is vertical velocity? The authors should state whether they are performing their particle tracking experiments using 2D or 3D velocity.

Response: [Page 6, line 25-26, in change-track-mode revised manuscript] In section 2.5, we insert two sentences, as

"w is the vertical velocity"

and

"we use the three dimensional velocity (U, V, w) to track the tracers,"

Page 6, line 10. The authors give the start and end time of the particle tracking experiments here, and not in the caption of Figure 6.

Response: [Page 7, line 1–2, in change-track-mode revised manuscript] Corrected.

Page 6, line 19. I do not understand what the authors mean when they say that the "wind stress curl highlights the strong seasonal variations". Also, wind stress curl is not shown in a figure.

Response: [Page 7, line 12, in change-track-mode revised manuscript] We deleted the sentence.

Page 6, line 27. The statement that warm-core eddies "seem to release footprints in the wind stress curl" does not make sense. I would have thought that the wind influences the eddies and not the other way around. Furthermore, when talking about wind stress curl, the authors refer to Figure 1 (b), but the figure caption says that this panel shows wind velocity and vorticity, not wind stress curl.

Response: [Page 7, line 20–21, in change-track-mode revised manuscript] We deleted the sentence.

Page 6, 29. It is not clear why the westward current is "remarkable". Has this feature not been observed before? Is it significant? Do the authors believe that it cannot be explained by their observations, or by existing theory?

Response: [Page 7, line 12, in change-track-mode revised manuscript] We noted the westward current "remarkable" according to the magnitude and zonal extent. Present observation displays the vertical structure of this westward current (west propagating planetary wave, WPPW).

We added discussion on this west propagating planetary wave in Section Discussion, as "From the theoretical viewpoint, the phase speed of first-, second- and third-mode baroclinic Rossby wave at 6°N in Indian Ocean is roughly 0.6, 0.2 and 0.1 m/s, respectively Subrahmanyam et al. (2001). The phase speed of WPPW match well with that of the second-mode baroclinic Rossby wave. For the generation mechanics, Subrahmanyam et al. (2001) argued that this kind Rossby wave was probably radiated from coastal trapped Kelvin wave at south-west coast of Indian. Meanwhile, this kind Rossby wave can bring wave energy from south-west coast of Indian to the Somali coast, and feed the Somali Current and Somali eddies. The present study displays the vertical structure of this kind Rossby wave, however, the dynamics of WPPW (and Rossby wave) and the its association with Somali Current and Somali eddies call for further study. "

Page 7, line 7. Strictly speaking, the boundary between the tropics and the subtropics is around 23.5°N, so all of the observations being considered here are from the tropics. Consequently, the use of the word subtropical is misleading. Subsequent uses of these terms should also be revised.

Response: We corrected the statement accordingly. We changed the "tropical band" to "equatorial band", and changed the "subtropical band" to "tropical band" in the whole paper.

Page 7, line 7. I do not think the authors are justified in grouping PGW and RSW as RSPGIW. In Figure 2, the water masses are observed to be separate, and the authors have acknowledged the different densities of the two water masses.

Response: [Page 9, line 13–14, in change-track-mode revised manuscript] We deleted the statements related to RSPGIW.

Page 7, line 18. The observations from the World Ocean Atlas presented in Figure 3 are from a climatology and should not, therefore, be described as a snapshot.

Response: [Page 9, line 15, in change-track-mode revised manuscript] We corrected the statement.

Page 7, line 19. The authors have not marked north and south on Figure 3, so it is difficult for the reader to make sense of statements such as "the thermocline... deepens northward". Response: [Page 19, Figure 4, in change-track-mode revised manuscript] We therefore added

the latitude in Figure 3 (Figure 4 in revised manuscript).

Page 7, line 22. I do not really agree with the authors's point about ventilation. The outcropping of isotherms visible in Figure 3 is from within the mixed layer? it is not clear that "subsurface" water is then able to take part in air-sea interaction.

Response: We argue that the mixed-layer is approximately well-mixed bulk layer, which can be defined as the layer of temperature within SST minus 0.1 (or 0.5) °C. In Figure 3 (Figure 4 in revised manuscript), the interval of temperature contour is 1.0 °C, therefore, we suppose that the outcropping isothermal line was below the mixed-layer, and argue the "subsurface" water taken part in air-sea interaction.

Page 7, line 28. The authors do not explain why their observations show that saline intermediate waters are PGW and RSW ? nor do they explain why this is not clear in the climatology. Response: [Page 9, line 28–29, in change-track-mode revised manuscript] Basically, we identified the PGW and RSW from T-S diagram. We added the reason in revised manuscript. For explain on the climatology, we added

"In other words, the comparison implies that the year 2002 is an anomaly year on the distributions of ASHSW, PGW and RSW."

Page 7, line 31. Why have the authors chosen the 22 kg m-3 isopycnal?

Response: We chosen 22 kg m-3 because it was the first near-surface isopycnal in climatology, meanwhile, and it characterized the differences among four datasets.

Page 7, line 33. It is hard to compare Figure 1b and Figure 3, because Figure 1b uses degrees and Figure 3 uses km.

Response: [Page 18, Figure 4, in change-track-mode revised manuscript] We added the latitude in the Figure.

Page 7, line 33. I do not understand what is meant by projecting the outcrop point in the wind vector field.

Response: [Page 9, line 32, in change-track-mode revised manuscript] We deleted "when we project the outcrop point in the wind vector field".

Page 8, Figure 2. I am not sure why the mean temperature-salinity curves have been plotted. They are not mentioned in the text and they distract from the new observations.

Response: [Page 10, Figure 3, in change-track-mode revised manuscript] We moved out the mean temperature-salinity curves.

Page 9, line 1. I don't think that the thermocline can be described as descending "sharply". Response: [Page 11, line 1, in change-track-mode revised manuscript] We deleted "sharply".

Page 9, line 5. I don't understand what is "remarkable" about the southward extension of salty water.

Response: [Page 11, line 6, in change-track-mode revised manuscript] "the remarkable southward extension of salty water in the upper ocean"

is modified to

"the southward extent of ASHSW".

Page 9, line 15. Again, I don't understand why the velocity field in the upper ocean is "re-markable".

Response: [Page 11, line 16, in change-track-mode revised manuscript] We changed "remarkable" to "relatively strong".

Page 9, line 19. I think the authors need to label features of interest? eg CCE2 ? on Figure 4. As it stands, it is quite hard to see what the authors want the reader to look at.

Response: [Page 20 Figure 5, in change-track-mode revised manuscript] We added the position of CCE1, CCE2 and WPPW in the Figure.

Page 9, line 22. The authors need to discuss, somewhere in the paper, the significance of the westward-propagating disturbance. At the end of the paper, I have no better idea of what is it and why it might be significant than I had at the start.

Response: [Page 2 line 24-28, in change-track-mode revised manuscript] In the introduction section, we added:

"Besides, the planetary waves at-least include Rossby and Kelvin waves (Rhines, 1975; Mc-Creary, 1985). Satellite-retrieved sea surface height are commonly used to detect the planetary waves (Chelton and Schlax, 1996), however, the internal dynamics of planetary wave is not sufficiently addressed. The phase speed of west-propagating planetary wave (WPPW) is used to match the theoretical Rossby wave, nonetheless, the vertical structure of WPPW calls for vertical profiling observation (Subrahmanyam et al., 2001)."

Page 9, line 26. When comparing the observationally derived geostrophic current and the current fields from the re-analysis products, have the authors considered that the current fields in the re-analysis products might contain ageostrophic components, eg Ekman flow?

Response: [Page 12 line 4–9, in change-track-mode revised manuscript] In section 3.3, we added a paragraph on discussion the Ekman flow:

"Part of the difference between observation-based absolute geostrophic current and reanalysis current is due to the near-surface Ekman current. The mean surface Ekman speed, which is approximately the difference between surface geostrophic current and in-situ surface current (from surface drifter), is within 0.1 m/s in northern IO (Saj, 2017). Besides, the climatological monthly mean mixed-layer depth in May is roughly 20 m at station 9.5°N and 59.5°E (on CR; Liu et al., 2018). Considering the near-surface Ekman current decays exponentially with depth, therefore, the near-surface Ekman currents in reanalysis datasets are relatively weak to affect the main results as mentioned above."

Page 10, first paragraph. I do not understand the argument that this paragraph is trying to make. There are several ideas that are not fully explored and which are insufficiently referenced. The authors seem to be contrasting "customary ventilation theory" and its corresponding meridional flow with "potential vorticity analysis" and its corresponding zonal flow. This strikes me as being a massive oversimplification; at the least, it requires a much more detailed explanation. Much of this material " as well as the extra explanation I would like to see added " probably belongs in the introduction.

Page 10, line 1. In what way are explanations of ASHSW and RSW pathways "ambiguous". This statement needs to be referenced, evidenced and more fully explained, either here or in the introduction.

Response: [Page 2 line 10-17, in change-track-mode revised manuscript] We moved the paragraph into the Introduction Section. We modified the explanation, as

"For instance, three water masses were defined in NWIO as Arabian Sea High-Salinity Water

(ASHSW), Persian Gulf Water (PGW) and Red Sea Water (RSW). Regarding the pathways of these water masses, the mechanics are not clear. RSW is formed near the northern side of the NWIO; therefore, according to the customary ocean ventilation theory, RSW sinks and moves southward along the isopycnal layer from the generation zone following the wind-driven current (Luyten et al., 1983). However, the applicability of the ocean ventilation theory is still unknown for the northern IO, because the surface wind reverses direction under the influence of winter monsoon (Liu et al., 2018). Meanwhile, the limited meridional extent of IO omits the polar-to-subpolar front, which helps form intermediate water in Pacific and Atlantic Oceans (You, 1998). In contrast, in situ potential vorticity analysis on RSW reveals that the flows generally follow the zonal direction (Beal et al., 2000). How the water mass moves is worthy further investigation."

Page 10, line 5. This sentence is poorly expressed.

Response: [Page 2 line 14-18, in change-track-mode revised manuscript] We moved the paragraph into the Introduction Section. We changed

"However, the feasibility of the ocean ventilation theory is still under debate, especially for the northern IO, whose meridional extent is limited compared with the other two basins." into

"However, the applicability of the ocean ventilation theory is still unknown for the northern IO, because the surface wind reverses direction under the influence of winter monsoon (Liu et al., 2018). Meanwhile, the limited meridional extent of IO omits the polar-to-subpolar front, which helps form intermediate water in Pacific and Atlantic Oceans (You, 1998). "

Page 10, second paragraph. The authors should explain clearly why they break up the particle tracking results into latitudinal bands? is this because they suspect different processes/currents are causing differences in circulation between these bands? Also, this paragraph should make some attempt to elucidate these processes and to explain what's new and important about these results. At present, the text just explains what the reader can already see in the figure. Page 10, line 14. I do not agree that this looks like flow in the summer Somali Current.

Response: [Page 12, line 21 – Page 13 line 2, in change-track-mode revised manuscript] We revised the paragraph. We added:

"For better describing the trajectories, we separate the CR to three latitude bands as 2.3-5°N (equatorial band), 5-8°N and 8-9.8°N."

"In the equatorial band (Fig. 6e), the near-equator tracers come from the west side, which is consistent with the north branch of East African Coastal Current during summer monsoon (Schott and McCreary Jr., 2001; Schott et al., 2009)."

"Meanwhile, for the relatively north-side tracers in equatorial band, the trajectories backtrack to east side, which is probably following a westward current or meso-scale eddy."

"For latitudes from 8-9.8°N (Fig. 6a), the trajectories emphasis the north branch of East

African Coastal Current, meanwhile, the water at the north station of CR comes from the northeast side, and one station water shows the cross-equatorial current around  $53^{\circ}E$  (east of Southern Gyre; Schott et al., 2009). "

Accordingly, we deleted:

"For the tropical band (Fig. 6e), the water mainly follows the zonal movement, but the nearequator tracers are from west side and relatively north-side tracers come from east side."

"For latitudes from 8-9.8°N (Fig. 6a), the trajectories look like the flow of the summer Somalia Current (Schott et al., 2009)"

Page 10, third paragraph. Again, some discussion of processes is needed here.

Response: [Page 13, line 3–15, in change-track-mode revised manuscript] We rewritten the paragraph. Original version:

For the RSW in the intermediate-depth layer at 700 m, the trajectories in the tropical band (Fig. 6f) and at latitudes from 5-8°N (Fig. 6d) generally follow the zonal movement (Beal et al., 2000). The tracer movements at latitudes from 8-9.8°N (Fig. 6b) partly agree with those of the ventilation theory, and partly follow the zonal direction (Beal et al., 2000). New version:

For the RSW in the intermediate-depth layer at 700 m, the trajectories in the equatorial band (Fig. 6f) show the zonal movement. Most of the trajectories move from the west side, which coincident with the potential vorticity explanation (Beal et al, 2000), that the RSW moves southward along the coast with the help of winter monsoon, and then leave the coast and shift to middle ocean via zonal movement. Other two trajectories come from east side, and one extra trajectory moves from northwest with circular track. Hence, these trajectories display three kinds of pathways. Accordingly, in the 5-8°N band (Fig. 6d), the mainly eastward zonal movements agree with (Beal et al., 2000), meanwhile, some westward trajectories resemble the ventilation theory (Luyten et al., 1983; Qiu and Huang, 1995). At last, in the 8-9.8°N (Fig. 6b), new pathway directly from northwest is emerged, and the trajectories support 700 m waters are probably directly from east of the Horn of Africa (or Gulf of Aden) without southward movements along the coast.

Page 10, line 25. Are there papers or technical reports available that explain methodological differences between the HYCOM and SODA re-analyses? Would more thorough research negate the need for speculation?

Page 10, line 27. It is not clear to me what is meant by the "dynamic core" of the ocean general circulation model.

Response: [Page 13, line 26– Page 14 line 4, in change-track-mode revised manuscript] We nearly rewritten the paragraph. We changed

"We speculate that although both SODA and HYCOM assimilate the Argo data into an Oceainc General Circulation Model (OGCM), the methodology of assimilation or the weight between OGCM and in situ observations, is sharply different. We assume the southward extension of ASHSW could be simulated by the dynamic core of OGCM, and the phenomenon was not captured in the Argo-only observation, therefore, HYCOM seems more to approach the dynamic model, and SODA weighted more on the Argo-only observations. Additionally, the finer horizontal resolution of HYCOM likely helps HYCOM involve more physically sound mechanics, such as the downwelling of salty water and wind-driven meridional movement." to

"Although both SODA and HYCOM assimilate the Argo data into Oceainc General Circulation Models (OGCMs), the assimilation methods of SODA and HYCOM are considerably different. SODA adopts optimal interpolation (Carton et al., 2008), while HYCOM uses 3D variational scheme. One advantage of 3D variational scheme versus optimal interpolation is the conservation of dynamical constraints (Zhu et al., 2006; Yin et al., 2012; Edwards et al., 2015). Therefore, HYCOM probably describes better on the wind-driven circulations, monsooninduced coastal current and meso-scale eddy movement, which are all related to the southward extension of ASHSW."

Page 10, line 33. I dislike the description of geostrophic current as an "alternative result for the ocean current". Geostrophic flow is an important part of ocean circulation and is perfectly valid in its own right.

Response: [Page 14, line 5-7, in change-track-mode revised manuscript] The sentences were deleted.

Page 11, conclusions section. The particle tracking results are not mentioned in the conclusion. Response: [Page 15, line 12–16, in change-track-mode revised manuscript] We added paragraph in conclusion section, as

"To explore the pathways of ASHSW and RSW during the expedition time, we set tracers in SODA dataset at depths of 100 and 700 m, and backtrack their trajectories via three dimensional Lagrangian description. Overall, for the 100 m depth waters, the results reveal the pathways related to the north branch of East African Coastal Current and the flow from northeast side (or Arabian Basin), while for the 700 m depth waters, the trajectories mainly follow the zonal direction from either west and east sides. The results give direct-viewing descriptions and call for further dynamical investigations."

Page 11, line 2. The authors have not discussed baroclinic modes in the results section, so it does not make any sense to the reader when the concept is introduced in the discussion section. Furthermore, are the authors certain that baroclinic mode is an appropriate concept in this instance?

Response: [Page 13, line 18, in change-track-mode revised manuscript] We changed the word "baroclinic mode" into "vertical structure".

Page 11, line 11. It is not correct to say that you integrate density to get the geostrophic current.

Response: [Page 14, line 32, in change-track-mode revised manuscript] We changed "we integrate the density field to obtain the absolute geostrophic current." into

"we compute the absolute geostrophic current based on the density profiles and sea surface height".

Page 11, line 20. I do not think that the authors have shown how "the present analysis shows potential data applications for the future".

Response: [Page 15, line 7–9, in change-track-mode revised manuscript] We changed

"The present analysis shows potential data applications for the future, where the meso-scale eddies are relatively important but cannot be well described by the Argo-only data source." into

"Over the NWIO, the meso-scale eddies are relatively important but cannot be well described by the Argo-only data source. The present analysis shows more data source for potential data assimilation experiment."

Page 14, Figure 3. Using km as the horizontal co-ordinate is unhelpful, given that in the text what is actually interesting is the location in degrees north. The authors should also mark on features of interest discussed in the text, such as the eddies.

Response: [Page 19, Figure 4, in change-track-mode revised manuscript] We added the latitude in the Figure.

Page 16, Figure 5. Given that the westward-propagating disturbance is so little discussed, I do not think that this figure adds anything to the paper.

Response: We removed this figure.

[revised manuscript text omitted]

---

## Author Response (AR3)

**Response to two Reviewers and Editor**

Reviewer 1

The authors have made substantial changes to the manuscript. I would suggest the authors make minor revisions to make the manuscript easier to read.

Response: We thank the reviewer for the valuable time and great efforts in evaluating our work. Detailed responses to the constructive comments are as follows.

1. Figure 2. "monthly wind verctor?". Monthly-climatological or May 2012?

Response: Yes. Monthly mean of May 2012. We noted this information in Figure 2.

2. Figure 4. What periods of data are used for SODA, HYCOM, and WOA?

Response: We used monthly mean data of May 2012. We added this information in Figure 4.

3. Could you highlight the 22 kg/m$^3$ isopycnal in Figure 4?

Response: We labeled the 22 kg/m$^3$ isopycnal in new Figure 4.

4. "comparison implies that the year 2002 is an anomaly year for the distributions of ASHSW, PGW and RSW." Could you give some interpretation here? Maybe you could add a sub-graph to show the wind anomaly in May 2012 in Fig. 2, if the conclusion "the ventilation is induced by surface wind" is correct.

Response: We therefore look into the inter-annual variation of wind anomaly. For the 7-year averaged monthly mean wind vorticity (Figure A1), a negative vorticity from wind is observed in Arabian Basin. Then in the year 2002 (Figure A2), the negative vorticity is further enhanced (negative anomaly) in Arabian Basin. In contrast, in some other years, it is relatively weak (positive anomaly). The negative wind vorticity input is the main reason of south movement of ASHSW (Schott et al., 2009; Sverdrup Balance). However, the description is not quantitative, we hope to do more work in the future.

Schott, F. A., Xie, S. P., and McCreary Jr. J. P.: Indian Ocean circulation and climate variability, Reviews of Geophyscis, 47, RG1002, 2009.

5. Too many abbreviations are not benefit for reading.

Response: We deleted the abbreviations including IO (Indian Ocean), IOD (Indian Ocean Dipole), RAMA (Research Moored Array for African-Asian-Australian Monsoon Analysis and Prediction), ADT ( absolute dynamic height), IEW (Indian Equatorial Water) and ASW (Arabian Sea Water).

6. Figure 6. As you discussed Fig. 6e first, and then Fig. 6c and Fig. 6a, you may want to rearrange the subgraphs.

Response: [Page 12-13, Change-Track-Mode revised manuscript] The sequence in Figure 6 is consistent with the geographic distribution. Therefore, we rearranged the text instead.

7. Please add some information about the "starting/ending point" in Figure 6 caption.

[Figure]

**2009-2015 Monthly Mean: May**

Figure A1: 7-year averaged monthly mean wind vorticity (2009-2015; May).

Response: Corrected.

8. "We set some passive tracers along the CR." Do you mean you set the traces along the CTD/XCTD stations shown in Fig. 1? If so, it would be helpful to show the section in Figure 6.

Response: We added the section in Figure 6.

9. To me, the title is not appropriate.

Response: Yes, the title looks general. But the time and space define the specificity of the study. We kept the title finally.

10. The revised "Introduction" is much better than the former. I noticed you raised several questions: pathways of water masses, and mechanics for their moves are not clear. The meso-scale eddies and planetary waves are not sufficiently observed. The internal dynamics of planetary wave is not sufficiently addressed. I also noticed you added several sentences to state the purpose/content of this manuscript: "First, hydrographic survey takes a snapshot on the water mass, and gives an evidence on the activity of water mass. Second,?". I feel like there is

[Figure]

Figure A2: Yearly anomaly relative to the 7-year averaged monthly mean wind vorticity (2009-2015; May).

a disconnect between the "existing questions" and the "answers". I thus kindly ask the authors to think about this issue more, in order to make the manuscript better.

Response: [Page 2, Line 3-4, Change-Track-Mode revised manuscript] According to the suggestions, we changed "Regarding the pathways of these water masses, the mechanics are not clear."

to

"Regarding the pathways of these water masses, the movements are not well observed, and the corresponding mechanics are not clear."

Reviewer 2

I thank the authors for revising the original draft of this paper, and for responding to my first review. This second draft is an improvement on the first – in particular the comparison of the observations and the re-analysis products – although there are a few points that I think require some further work.

Response: We sincerely thank the reviewer for bringing up many vital points to help us improving the manuscript.

I agree with the authors' point, made in their response, that the hydrographic observations are useful to our understanding of water masses of the Indian Ocean. But the value of these observations, as currently presented, is still not as clear as it could be. I think that this is because the literature the authors have cited is not the most helpful or relevant. I have taken a quick look at the Lutyen et al (1983) paper, and it appears to be too general to be the sole reference for the spreading of RSW (page 2, line 4). Papers specifically on the water masses of the NW Indian Ocean should be cited. I would encourage the authors to read and consider the following:

Beal et al, 2000. Spreading of Red Sea overflow water in the Indian Ocean. JGR, 105, 8549–8564

Bower et al, 2000. Character and dynamics of the Red Sea and Persian Gulf outflows. JGR, 105, 6387–6414

Durgadoo et al, 2017. Indian Ocean sources of Agulhas leakage. JGR: Oceans, 122, 3481–3499

Han & McCreary, 2001. Modelling salinity distributions in the Indian Ocean. JGR, 106, 859–877

Prasad et al, 2001. Seasonal spreading of the Persian Gulf Water mass in the Arabian Sea. JGR, 106, 17059–17071

Prasad & Ikeda, 2002. A numerical study of the seasonal variability if Arabian Sea High-Salinity Water. JGR, 107, 3197

Prasanna Kumar & Prasad, 1999. Formation and spreading of Arabian Sea High-Salinity water mass. JGR, 104, 1455–1464

Shapiro & Meschanov, 1991. Distribution and spreading of Red Sea Water and salt lens formation in the northwest Indian Ocean. 38, 21–34.

These papers should help to better place the authors' valuable observations – for instance, the southward extension of ASHSW – into an informative context, both in the introduction and the discussion. As it stands, the main conclusion regarding water masses is that ASHSW is found a little further south in the present observations than in a climatology: I don't think that this is enough.

Response: [Page 2, Line 1-29, Change-Track-Mode revised manuscript] We read these good references, and tried to digest these informations. We added more reviewing text in the Introduction Section.

Old version:

To date, the main water masses in the IO and NWIO have been described by the scientific community (Sharma et al., 1978, Kumar and Prasad, 1999; Emery, 2001; Talley et al., 2011). For instance, three water masses were defined in NWIO as Arabian Sea High-Salinity Water (ASHSW), Persian Gulf Water (PGW) and Red Sea Water (RSW). Regarding the pathways of these water masses, the mechanics are not clear. RSW is formed near the northern side of the NWIO; therefore, according to the customary ocean ventilation theory, RSW sinks and moves southward along the isopycnal layer from the generation zone following the wind-driven

current (Luyten et al., 1983). However, the applicability of the ocean ventilation theory is still unknown for the northern IO, because the surface wind reverses direction under the influence of winter monsoon (Liu et al., 2018). Meanwhile, the limited meridional extent of Indian Ocean omits the polar-to-subpolar front, which helps forming intermediate water in Pacific and Atlantic Oceans (You, 1998). In contrast, in situ potential vorticity analysis on RSW reveals that the flows generally follow the zonal direction (Beal et al., 2000). How the water mass moves is worthy further investigation. In contrast, in situ potential vorticity analysis on RSW reveals that the flows generally follow the zonal direction (Beal et al., 2000). How the water mass moves is worthy further investigation.

New version:

To date, the main water masses in the NWIO have been described by the scientific community (Sharma et al., 1978, Kumar and Prasad, 1999; Emery, 2001; Talley et al., 2011). For instance, three water masses were defined in NWIO as Arabian Sea High-Salinity Water (ASHSW), Persian Gulf Water (PGW) and Red Sea Water (RSW). The formations of ASHSW, PGW and RSW are all due to the high evaporation (Shapiro and Meschanov, 1991; Kumar and Prasad, 1999; Bower et al., 2000; Prasad et al., 2001; Prasad and Ikeda, 2002). Regarding the pathways of these water masses, the movements are not well observed, and the corresponding mechanics are not clear. Kumar and Prasad (1999) described the climatological seasonal distribution of ASHSW using in-situ temperature and salinity fields. In the northern Arabian Sea, ASHSW forms in the surface during winter, and moves southward due to the surface wind. Otherwise, the multi-scale variations of ASHSW were not sufficiently documented (Kumar and Prasad, 1999; Prasad and Ikeda, 2000). According to the customary ocean ventilation theory, ASHSW sinks and moves southward along the isopycnal layer from the generation zone following the wind-driven current (Luyten et al., 1983). However, the applicability of the ocean ventilation theory is still unknown for the NWIO, because the surface wind reverses direction under the influence of winter monsoon (Liu et al., 2018). RSW supplies important intermediate water salinity source in the entire Indian Ocean basin (Han and McCreary Jr., 2001). Formation and spreading of RSW exhibit seasonal variations (Bower et al., 2000; Beal et al., 2000). Based on the long-term hydrographic data, the occurrence of RSW show four possible branches of RSW around the Gulf of Aden: First flows eastward to the Arabian Basin, the second moves southward to Somali Basin, the third spreads southward along the Somali coast, and the fourth moves northeastward along the Arabian coast (Shapiro and Meschanov, 1991). Later, Beal et al., (2000) highlighted the branch along the Somali coast, meanwhile, the potential vorticity analysis revealed that the flows generally followed the zonal direction in NWIO. It is noted that the mechanics of intermediate water in Indian Ocean should be different from that in the Pacific and Atlantic Oceans. The limited meridional extent of Indian Ocean omits the polar-to-subpolar front, which helps forming intermediate water in Pacific and Atlantic Oceans (You, 1998). How the RSW moves is worthy further investigation (Durgadoo et al., 2017).

Besides, we cited references in the Section Results (Section 3.2 and 3.4), including Kumar and Prasad (1999), and Shapiro and Meschanov (1991).

The scientific message of the paper would be improved further if the particle tracking were better integrated with the discussion of water masses. The final paragraph of Section 3.4 still feels a little brief. As such, the scientific value of the particle tracking is not clear. Again, I would encourage the authors to discuss the results of their experiments in the context of work specific to the NW Indian Ocean, and the papers listed above should also be helpful here. Trajectories in the PGW and RSW layers are potentially very informative given the distance between the Carlsberg Ridge and the Persian Gulf/Red Sea. Plotting the temperature/salinity properties of particles at the time and location of initiation might also be very informative.

Response: We tried to show Lagrangian trajectories of water masses based on ocean reanalysis data. However, the motivation of present work is to display the onetime hydrographic survey. The systematic work on Lagrangian descriptions of PGW and RSW beyonds the main scope of present work, and leaves for future work. We tried to add the temperature information in Figure 6 (Figure A3), while the temperature information affect the visualisation of trajectory (i.e. Figure A3c–d). We finally did not add the temperature information.

[Figure]

Figure A3: Trajectories of tracers, the water temperatures were added.

Regarding the interpretation of the particle trajectories, I am not convinced that the westward trajectories visible in Figure 6(e) are the result of eddy transport – surely the trajectories are too

straight for this to be the case? Given the location of this westward flow and the SODA velocity distribution presented in Figure 5, is it possible that these trajectories represent transport by the WPPW?

Response: [Page 12 Line 32, Change-Track-Mode revised manuscript] We added the possible of WPPW in the hypothesis. We believe there are a lot information behind these trajectories, and the dynamics is rarely discussed here. We hope to do more research in the future.

The methods section is much clearer than in the previous draft. Now that I understand what the authors have done, I have a couple of points to raise. Firstly, I am not convinced that no calibration is required. The differences between the CTD and XCTD are, at depth, not insignificant. If the variance about the quoted mean difference in the intermediate ocean is not excessively large, I would have thought that calibration of the XCTDs against the ship's CTD would be necessary. Secondly, if Argo observations are already processed and calibrated by the Argo programme prior to being made available " and given that the distance between the Argo and CTD profiles is almost twice that between the XCTD and CTD profiles " I would have thought that this comparison could be removed: it doesn't add anything.

Response: [Page 3 Line 32-33, and Page 4 Line 1-9; Change-Track-Mode revised manuscript] We deleted the comparison between Argo and CTD. On the other aspect, for the comparison between CTD and XCTD, because the XCTD sensors are expandable, it is probably not convincing to use one XCTD error to calibrate all other XCTD. We therefore keep the other post-processing method unchanged.

Finally, I remain puzzled by the discussion about ventilation (page 8, final paragraph). Firstly, the definition of the mixed layer given in the authors' response to my first review should be included in the paper, as it is relevant to the discussion. Secondly, the authors state that north of the outcropped 22.0 kg $m^{-3}$ isopycnal, downwelling is likely occurring. But I cannot see how downwelling would produce the upward doming of the near-surface isopycnal at this location. Have the authors considered that this could be a boundary between high-salinity ASHSW in the centre of CCE1 (northern limit of the section – and it's a cold-core eddy, so we would expect upwelling in the centre?) and fresher surface water outside of the eddy (i.e. to the south)?

Response: [Page 9 Line 14-15; Change-Track-Mode revised manuscript] The definition of thermocline was adopted from Xie et al. (2002), as the 20°C isothermal line. Regarding the isopycnal layer, we simply matched the density structure as the multi-layer model, where the $22 - 22.5$ kg m$^{-3}$ layer exposed to the surface, and took part in the air-sea interaction (Figure 4; density structure).

Editorial comments as previously sent to improve clarity.

Page 2.

Line 15-16. "The phase speed of west-propagating planetary waves (WPPW) is used to match the theoretical Rossby wave". This sentence is unclear. I think not "is used". Does it refer to previous work (give a reference) or to the present manuscript (move it to a better place)?

Response: [Page 3 Line 2-3; Change-Track-Mode revised manuscript] We added two references

as "(Chelton and Schlax, 1996; Subrahmanyam et al., 2001)".

Lines 19-20. "interdisciplinary survey" (spellings).
Response: [Page 3 Line 6-7; Change-Track-Mode revised manuscript] Corrected.

Line 21. Not "activity". Maybe "movement" or "distribution".
Response: [Page 3 Line 8; Change-Track-Mode revised manuscript] Corrected.

Line 31. "All stations were mainly located". Either "all" or "mainly" (which means most but not all) ? but not both! Perhaps you mean "located close to the CR section".
Response: [Page 3 Line 18; Change-Track-Mode revised manuscript] Corrected.

Page 4 line 3. Better ". . stations in these comparisons . ."
Response: [Page 4 Line 7; Change-Track-Mode revised manuscript] Corrected. Later the sentence was deleted.

Page 6 Line 19. Better omit "sufficient" (what is the upwelling sufficient for?).
Response: [Page 7 Line 14; Change-Track-Mode revised manuscript] Corrected.

Line 28. "which refers to the" → "with respect to its"
Response: [Page 7 Line 23; Change-Track-Mode revised manuscript] Corrected.

Page 7 Near bottom "current is -0.38 m/s. . ." Why "-"? (presumably because westward). A magnitude must be positive.Your response to reviewer 2 refers to Maximenko et al. (2005). You should cite it here.
Response: [Page 7 Line 25-27; Change-Track-Mode revised manuscript] The sign of current magnitude was corrected. The reference was changed to "Subrahmanyam et al. (2001)". The added the citation of Subrahmanyam et al. (2001) here.

Page 8 Line 16. "and show a clear" → "suggesting a". Not "clear" because only one isotherm.
Response: [Page 9 Line 18; Change-Track-Mode revised manuscript] Corrected.

Line 27. Better "When we move forward to" → "In"
Response: [Page 9 Line 29; Change-Track-Mode revised manuscript] Corrected.

Page 10 lines 19-20. Please explain "observation-based absolute geostrophic current". Maybe you need a subsection 2.n about this. I see you have a sentence in the conclusions ? too late!
Response: [Page 11 Line 11; Change-Track-Mode revised manuscript] At the beginning of this section, "The geostrophic current" was changed to "The observation-based absolute geostrophic current".

Page 11 line 28. "The CTD and XCTD data are precise in reconstructing the three-dimensional oceanic data". This is a bold statement! The second horizontal dimension (x) is hardly measured except at the surface. So there have to be assumptions about the profiles away from the CR section.

Response: [Page 13 Line 19; Change-Track-Mode revised manuscript] Corrected. "precise" was changed to "useful".

Page 12 lines 25-29. The meaning of this sentence is unclear because the grammar is wrong. Please break it up, decide what is the most important part, give every clause a verb.

Response: [Page 14 Line 16-22; Change-Track-Mode revised manuscript] Corrected.

"Because present paper concentrates on the CR as well as the cross-ridge current, the upper 1050 m dynamics towards the upper ocean cross-ridge water transport, which is related to the deeper ocean dynamics through pressure adjustment, meanwhile, the activity of meso-scale eddy induces deeper ocean vorticity response (Rhines and Young, 1982), 
[revised manuscript text omitted]
. ~~Because present paper concentrates on the CR as well as the cross-ridge current, the upper 1050 m dynamics towards the upper ocean cross-ridge water transport, which is related to the deeper ocean dynamics through pressure adjustment, meanwhile, the activity of meso-scale eddy induces deeper ocean vorticity response (Rhines and Young, 1982), 
[revised manuscript text omitted]

---

## Author Response (AR4)

**Response to Editor**

Regarding Referee 1: title. Please think again. Yes, your title is specific enough, but what readers want to know is not that you did a hydrographic survey but what you learned from it. The title should reflect the new science, e.g. "Anomalous distribution and movement of distinctive water masses in the north-west Indian Ocean, May 2012". But don't just copy this, choose your own title.

Response: We changed the title to "Anomalous distribution of distinctive water masses over the Carlsberg Ridge in May 2012 ".

Regarding referee 2: what is the benefit from the particle tracking with SODA and HYCOM since you say that their currents are wrong? You need to discuss ? probably in section 4 ? how similar or different are the water masses suggested by the particle tracking compared with the hydrography. Are the differences consistent with the error in the reanalysis currents (and hence particle tracks)?

Response: In order to bridge the gap between the in-situ hydrographic study and SODA-based particle tracking, we added a paragraph in the "Section Discussion", as

"Present study uses SODA reanalysis to investigate the origins of water particles over the CR. However, the corresponding results need further validation. For instance, present study reveals both the meso-scale eddy and WPPW are misinterpreted by SODA, therefore, the waters trapped in meso-scale eddy and WPPW probably move in wrong ways. Meanwhile, trajectories from different oceanic reanalysis are probably different, regarding the south extents of ASHSW are not same in SODA and HYCOM. "

Here are some more detailed comments. Page 2

Lines 4, 17. Better "mechanics" → "dynamics"

Response: Corrected.

Lines 13-15. These seem to be in random order. Go clockwise from one coast to the other?

Response: Corrected. We rotated anti-clockwise from Somali coast to Arabian coast.

Line 27. "(WPPW) is used to match the" → "(WPPW) has been matched to the"? Anyway, if you are describing previous work then not "is" but "has been".

Response: Corrected. Thanks for the suggestion.

Page 3 Lines 1-2. "clear objective" (what is it?) and "analyze" (for what?). This is the place to say explicitly what are the objectives! You could repeat the three "reasons" in the previous three sentences. Or better re-write all four sentences so that you emphasise the science to be gained and not the methods. Methods are described in section 2.

Regarding "clear objective", we changed

"Hence, this paper aims to analyze hydrographic information by combining both CR expedition and Argo floats."

to

"Hence, this paper is motivated by observing the water masses over CR, describing the vertical structures of meso-scale eddy and planetary wave, and comparing the results of widely-used oceanic reanalysis in the NWIO."

Line 19. "instrument" → "different instruments"
Response: Corrected.

Page 7 lines 7-12. In a previous response you said "As refer to Maximenko et al. (2005), present observation displays the vertical structure of this westward current. The related theory needs to be further confirmed. Maximenko, N. A., Bang, B., and Sasaki, H.: Observational evidence of alternating zonal jets in the world ocean, Geophysical Research Letters, 32, L12607, 2005." I then said that you should include this and reference to Maximenko et al. (2005) in your manuscript. You have not! However, I am not sure what is the point of your previous response. If your previous response is saying that now you show the vertical structure of the westward current, and this is new (and certainly an improvement on Maximenko et al. 2005), then you should say this in your manuscript. However, that may be most relevant at page 10, line 27.
Response: In the early time, we used Maximenko et al. (2005) to emphasize the importance of profiling observation on west-propagating disturbance. During that time, we used a relatively general concept as "disturbance" to describe the "broad-longitude and narrow-latitude signal". We later changed the concept "disturbance" to "planetary wave", to exhibit the wave-like evolution. We also changed the reference from Maximenko et al. (2005) to Subrahmanyam et al. (2001), considering Subrahmanyam et al. (2001) systematically studied the SSH-based Rossby waves in the Indian Ocean. Due to the sparse observation of Argo, the vertical structure of second-mode Rossby wave is rarely observed by Argo-only dataset. More important, present paper shows the discrepancy of oceanic reanalysis in reconstructing the vertical structure of WPPW. Theoretical Rossby wave is obtained by vertical normal mode expansion, and the horizontal propagation of Rossby wave is nearly only along the zonal direction. Therefore, better observation of Rossby wave needs a standard meridional section, and we leave the dynamics of WPPW (or second-mode Rossby wave) for future study.

Page 8, figure 2 caption, end. "consistent surface geostrophic current"; do you mean "corresponding surface geostrophic current" meaning that it is consistent with the surface height?
Response: Yes. We changed "consistent" to "corresponding".

Page 10 Line 1. "2002" → "2012"?
Response: Corrected. We sorry for the mistake.

Line 10. "crops out" → "outcrops" or "surfaces".
Response: Corrected.

Lines 17-18. Regarding referee 2: "Finally, I remain puzzled by the discussion about ventilation . . the authors state that north of the outcropped 22.0 kg m$^{-3}$ isopycnal, downwelling

is likely occurring. But I cannot see how downwelling would produce the upward doming of the near-surface isopycnal at this location. Have the authors considered that this could be a boundary between high-salinity ASHSW in the centre of CCE1 (northern limit of the section ? and it?s a cold-core eddy, so we would expect upwelling in the centre?) and fresher surface water outside of the eddy (i.e. to the south)?? Your response: ". . . Regarding the isopycnal layer, we simply matched the density structure as the multi-layer model, where the 22 - 22.5 kg m$^{-3}$ layer exposed to the surface, and took part in the air-sea interaction (Figure 4; density structure)." I do not understand the relation of your response to the question. And I am puzzled by your present text "The upper ocean density fields from SODA and HYCOM also show clear ventilation structures. From the observations, equatorial waters with a potential density of 22 kg/m$^3$ at a depth of 30 m are rising to the surface." Where are the "ventilation structures" and why "ventilation"?

Response: Sorry for the mistake. The "downwelling" should be corrected to "downward movement". We tried to explain that the negative wind vorticity is downwelling-prefer wind forcing. In fact, negative wind vorticity induces downward movement near the surface, and the surface water can easier sink to lower layer. Here, in the outcropping multi-layer model, when the downwelling-prefer wind forcing exists on the surface, the ocean surface water sinks, other than the occurrence of upward/downward doming of near-surface isopycnal. Due to this reason, we emphasized the outcropping multi-layer model in the previous response. As a revision, we changed

"The north side of the outcrop point has negative wind vorticity, which promotes downwelling."
to
"The north side of the outcrop point has negative wind vorticity, which promotes downward movement."
Meanwhile,
"Ventilation is highly related to the downwelling of high-salinity water and its southern extent ... "
was changed to
"Ventilation is highly related to the sinking of high-salinity water and its southern extent ...."
On the other hand, we think the "ventilation structure" means outcropping multi-layer model, where the second layer (third layer, ... , or $N$th layer) has a chance to lift-up to surface, and take part in the air-sea interaction. In contrast, for the typical multi-layer model, there is always a mixed-layer in the surface, and the second layer could not be exposed to the atmosphere. For further explain the "ventilation structure", we added a sentence as
"that the south-side subsurface layers are exposed to the surface in the north-side".

Line 27. Not "well rebuilt". Maybe "clear" or perhaps "well represented" but if you say "well" you need to state the basis on which you claim good quality.
Response: Corrected. We changed "well rebuilt" to "represented".

Line 34. Surely ". . upper 200 m, where the current speed . ."?

Response: Corrected.

Page 11 Line 2. "and merges with the WPPW. . ."
Response: Corrected.

Lines 22-23. "The results support ASHSWs are mainly from the Arabian Basin . ." This is not meaningful ? where else would ASHSW come from? I think you want "The results show water coming mainly from the Arabian Basin . ."
Response: Corrected.

Line 23. "into CR. In" → "onto the CR. On"
Response: Corrected.

Line 25. "start at the east" → "starting east"
Response: Corrected.

Line 26. Delete "at the"
Response: Corrected.

Line 33. ". . Meanwhile, pathway directly from northwest is emerged, and the trajectories support 700 m waters are". I think you mean ". . Meanwhile, there are pathways directly from the northwest and other trajectories show 700 m waters coming"
Response: We changed "pathway directly from northwest is emerged, and the trajectories support 700 m waters are directly from east of the Horn of Africa (or Gulf of Aden) ....."
into
"there are pathways directly from northwest, and these trajectories support 700 m waters are probably directly from east of the Horn of Africa (or Gulf of Aden) ...."

Page 12 Line 2. Better ". . westward trajectories resemble the prediction of ventilation theory . ." . However, if you say this you need to say more about what currents are expected from ventilation theory and why.
Response: We therefore added a sentence to explain the ventilation theory, as
"According to the ventilation theory, if the wind in the north boundary of NWIO was eastward, and the ocean density field had ventilation structure, then the flow in CR was southwestward, and the waters in CR moved from northeast side."

Line 11. "thus" → "because they"
Response: We changed "thus" into "and they".

Line 31. "Indian" → "India"
Response: Corrected.

Line 33. Delete "the"
Response: Corrected.

Page 13 Line 11. "display clear ventilation structures". You can only say this if you answer the points above regarding page 10 lines 17-18 and page 12 line 2.

Response: Yes. In the revised manuscript, we explained the ventilation structure in the results of density field, and we addressed the ventilation theory in "Section 3.4 Tracers".

Additionally, in the first paragraph of "Section 3.4 Tracers", we adjusted

[revised manuscript text omitted]